



# Assessing different imaging velocimetry techniques to measure shallow runoff velocities during rain events using an urban drainage physical model

Juan Naves[1], Juan T. García[2], Jerónimo Puertas[1], Joaquín Suárez[1], Jose Anta[1]

[1]Universidade da Coruña, Water and Environmental Engineering Research Team (GEAMA), Civil Engineering School, A Coruña, 15071, Spain
[2]Mining and Civil Engineering Department, Universidad Politécnica de Cartagena, Cartagena, 30203, Spain

*Correspondence to*: Juan Naves (juan.naves@udc.es)

**Abstract.** Although surface velocities are key in the calibration of physically based urban drainage models, the shallow water depths developed during non-extreme precipitations and the potential risks during flood events limit the availability of this type of data in urban catchments. In this context, imaging velocimetry techniques are being investigated as suitable non-intrusive methods to estimate runoff velocities, when the possible influence of rain has yet to be analyzed. This study carried out a comparative assessment of different seeded and unseeded imaging velocimetry techniques: Large Scale Particle Image Velocimetry (LSPIV); Surface Structure Image Velocimetry (SSIV); and Bubble Image Velocimetry (BIV), through six realistic but laboratory-controlled experiments where the runoff generated by three different rain intensities was recorded. First, the use of naturally-generated bubbles and water shadows and glares as tracers allows the unseeded techniques (SSIV and BIV) to measure extremely shallow flows, but these are more affected by raindrop impacts, which even lead to erroneous velocities in the case of the highest rain intensities. At the same time, better results were obtained with techniques that use artificial particles for high intensities and in complex flows. Finally, the study highlights the feasibility of these imaging techniques to be used in measuring surface velocities in real field applications and the importance of considering rain properties to interpret and assess the results obtained.

## 1 Introduction

Since the last years of the 19th century, urban drainage systems have fulfilled a fundamental mission that has enabled us to guarantee the hygienic-sanitary conditions and growth of denser cities (Butler et al., 2018; Brown et al., 2009). Specifically, the sustainability and flood response capacity of urban drainage systems are today threatened by several factors. These are led by the increment in impervious areas, due to urbanization (Shuster et al., 2005; Yao et al., 2016) and ongoing climate change, resulting in a higher and more frequent number of heavy rainfall events (Willems et al., 2012; Arnbjerg-Nielsen et al., 2013). The increased flood risk is a consequence of these factors (Chen et al., 2015) which must be accurately assessed (Apel et al., 2004; Martinez-Gomariz et al., 2016). This continuous development of impervious areas also leads to a significant increase in



the load and peak concentrations of pollutants, which are accumulated on urban catchments surfaces and can be washed off
and transported by runoff into drainage systems and eventually to aquatic media (Lee and Bank 2000, Anta et al., 2006; Zafra
et al., 2017; Muthusamy et al., 2018). This process depends on multiple factors (Hatt et al., 2004; García et al., 2017) and
requires a clear understanding of the surface drainage in urban areas from the hydrodynamic point of view.

In this context, physically-based urban drainage models can help to assess complex cases, such as the definition of the inlet
capacity of different storm drains to transfer runoff stormwater into sewers (Martins et al., 2018; Rubinato et al., 2018) or to
assess particle wash-off processes (Hong et al., 2016; Naves et al., 2020a). A precise characterization of the surface velocities
and flow depths is required when calibrating these models due to their key role in flood risk assessment and in the detachment
and transport of surface pollutants. However, punctual velocity measurement equipment such as Acoustic Doppler Velocimetry
(ADV) does not allow a two-dimensional velocity field to be obtained in large urban areas during flood events without
requiring a huge deployment of instrumentation and at great risk to workers. Furthermore, due to the shallow runoff flows
during non-extreme events, ADV reliability is reduced as it is an intrusive technique that also needs about 5-7 cm to obtain
velocity measurements (Cea et al., 2007). Imaging techniques are thus expanding in open and large-scale environments as non-
intrusive methods for the characterization of surface velocity fields (Aberle et al., 2017). Large-Scale Particle Image
Velocimetry (LSPIV) is an image velocimetry technique that provides velocity fields in large areas, even in the proximity of
hydraulic structures (Muste et al., 2008; Fujita et al., 1998; Kantoush et al. 2011). LSPIV velocity determination can be affected
by deficient illumination, diffuse light reflections or free-surface waviness generated by wind or large-scale turbulence
structures. Seeding is also a key parameter of the technique that may need to be artificially improved (Aberle et al., 2017). For
instance, the differences achieved in a full-scale stormwater detention basin, compared with the reference values, were below
14% although in some bordering points these could rise up to 44% (Zhu et al., 2019).

In addition, other imaging techniques may be applied to determine runoff velocities without the presence of particles. Surface
Structure Image Velocimetry (SSIV) is a LSPIV-based method which introduces some improvements based on image pre-
processing analysis (Lüthi et al., 2014; Hansen et al., 2017; Leitão et al., 2018). Hence, shadows and immovable objects
detected from a set of images are removed by subtracting the averaged image to the temporal series, and the structures presented
on free-flow surfaces such as the water reflections are enhanced. In addition, bubbles are used as tracers to estimate overland
velocities in the technique known as bubble imaging velocimetry (BIV). The BIV technique was first introduced to measure
the velocity field in high aerated flows from backlit images analysis without the need for laser-like illumination (Ryu et al.,
2005). Bubbly flows are illuminated by a uniform light source while a high-speed camera captures shadow textures created by
gas-liquid inter-faces (Aberle et al., 2017). Lin et al. (2012) have already used this technique to measure the flow structure in
hydraulic jumps in the aerated zone.

In Naves et al. (2019a), a variation of the LSPIV technique was applied to measure the surface velocity fields generated by
three different rain intensities in a full-scale urban drainage physical model. That study used UV illumination and fluorescent
particles as artificial tracers to satisfactorily address the problems caused by the presence of raindrops in the experiments,
which are the interference of raindrops in the visualization of images and the disturbances generated in the flow because of





raindrop impacts. To the best of the authors' knowledge, that was the first and only study where an imaging velocimetry

technique has been applied during rainy conditions. Despite the good results achieved and the great suitability for laboratory

applications of the proposed methodology, its transferability to field studies is restricted by the difficulties in using artificial

particles and special illumination. However, in addition to the interferences mentioned above, the raindrop impacts also

generate bubbles and some other structures on free-flow surfaces that may be used as tracers by the SSIV and BIV techniques.

Due to the great potential of these unseeded techniques to obtain overland flow velocity data in field applications using, for

example, pre-installed surveillance cameras (Leitão et al., 2018), studying their performance under rainy conditions is an

interesting and novel research gap to be addressed.

Therefore, experimental videos of the overland flow generated by three different rain intensities, under laboratory-controlled

conditions and recorded with and without artificial particles, are used in this study to comparatively assess the performance of

different seeded and unseeded imaging velocimetry techniques under rainy conditions. First, the sensitivity of the velocity

results to the analysis parameters is investigated in order to test the robustness of each method. Then, the resulting velocity

fields are compared in order to analyze the feasibility of using each technique in different characteristic flows developed in

urban catchments and to investigate the influence of rain intensity in velocity measurements. Finally, the potential usability of

these imaging techniques measuring surface velocities in real field applications is discussed.

## 2 Materials and methods

The experimental work performed to record the overland shallow flows generated by three different simulated rainfalls in an

urban drainage physical model is introduced first, in Sect. 2.1. Then Sect. 2.2 includes a description of the procedure followed

to obtain velocity results from the original video frames. Sect. 2.3 describes the strategy to assess the performance of different

image velocimetry techniques depending on rain intensity and the typology of flow. Finally, the surface areas where the

analysis was focused, the ranges of variation of the parameters involved in the assessment of the robustness of each technique,

and the procedure implementation details are explained in detail in Sect. 2.4, 2.5, and 2.6.

### 2.1 Experimental data

The experimental dataset described in Naves et al. (2020b) and freely available at Naves et al. (2019b) was used in this study

for the assessment of different imaging velocimetry techniques. The dataset comprises a series of videos where the surface

runoff was recorded in a 36 m$^2$ urban drainage physical model. The facility (Fig. 1a) consists of a full-scale street section

where the rainfall-runoff generated by a dripper-based rainfall simulator, which is able to produce three different rain intensities

(30, 50 and 80 mm h$^{-1}$), drains into a pipe system through two gully pots. Two types of configurations have been used to

visualize overland flow: a) experiments using fluorescent particles and UV illumination; and b) using white-LED lamps

without artificial particles to highlight air bubbles and water reflections generated by raindrops in the flow. While seeded

videos were already used in the application of a modified LSPIV technique in Naves et al. (2019a) as stated in the introduction



section, unseeded videos are used for the first time in this work to consider SSIV and BIV imaging velocimetry techniques. Figure 1b shows a scheme of the configuration of the experiments where the videos were recorded.

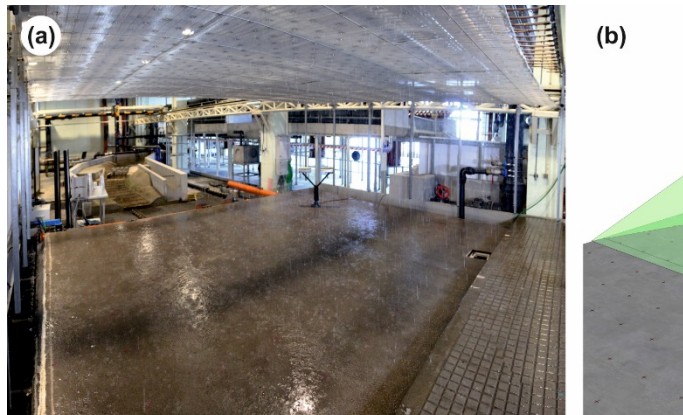
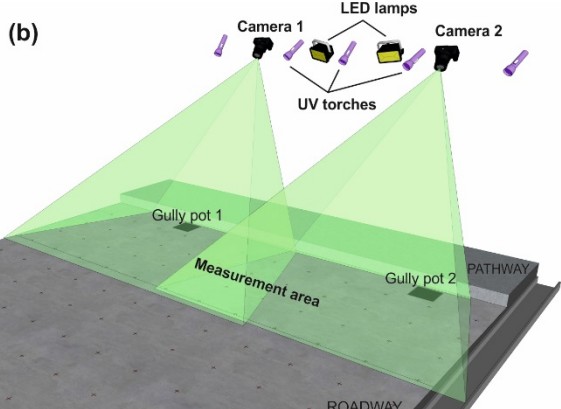

**Figure 1: General image of the urban drainage physical model with the rainfall simulator (a) and experimental setup of the PIV experiments (b).**

As seen in Fig. 1b, two Lumix GH4 cameras with 28 mm focal length recorded the first 2 m attached to the curb along 5 m of the physical model. UV torches and LED lamps were installed next to the cameras 2.2 m above the pathway. Once 4K resolution and 25 Hz videos were recorded, 1500 frames of steady flow (equivalent to 60 s) were extracted. The frames extracted were scaled and ortho-rectified using the known 2D coordinates of 28 and 24 reference surface points for each camera. Finally, the reference points placed in the intersection between the recorded areas of each camera were used to crop

and join the image, resulting in raw images where 1 pixel corresponds to 1 mm in real-world coordinates. Examples of these images obtained from UV seeded and LED unseeded experiments, which are available for others to use (Naves et al., 2019b), are included in Fig. 2. These six sets of images considering both the experimental setup and the three rain intensities were used as the basis for the different imaging velocimetry techniques assessed in this study. A more detailed description of the physical model, the simulated rain, or the procedure to extract the images can be consulted in Naves et al. (2020b, 2020c, 2019a),

respectively.





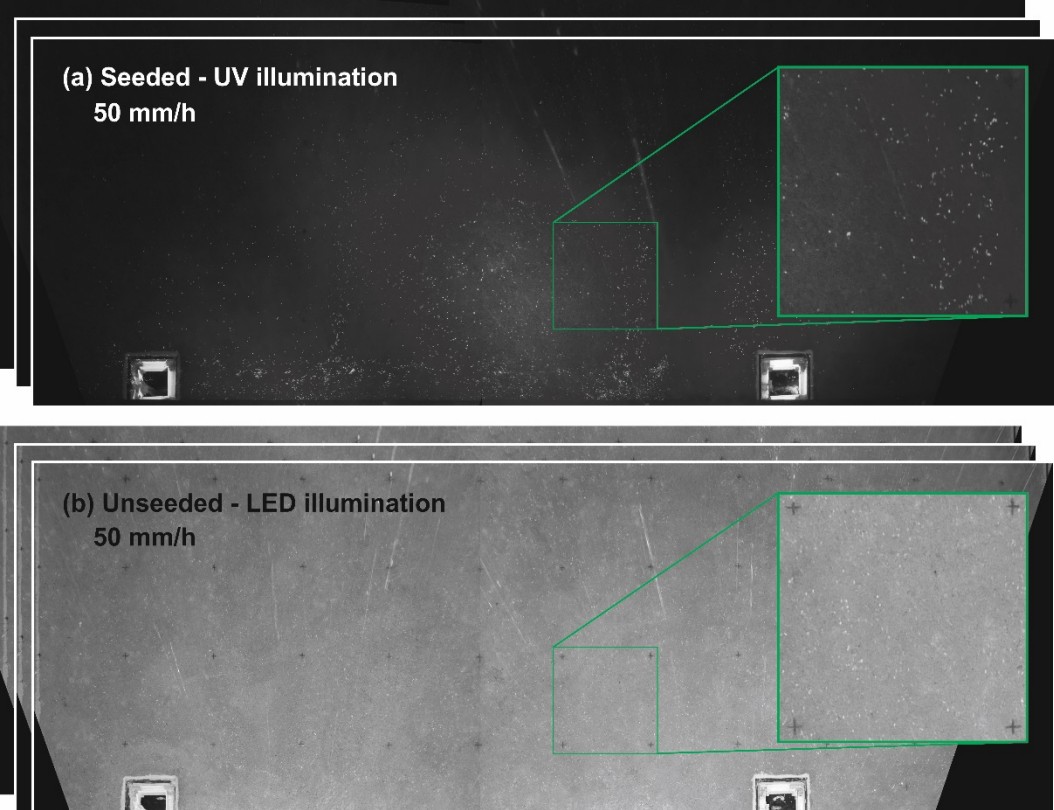

**Figure 2: Example of frames used for the assessment of different imaging velocimetry techniques. Images correspond with the two experiments performed for the rain intensity of 50 mm h⁻¹: using fluorescent particles and UV illumination (a) and using led lamps to highlight water reflections and air bubbles present in the flow (b).**

### 2.2 Analysis procedure and imaging velocimetry techniques

Four imaging velocimetry techniques were considered in this study to obtain the overland flow velocities from the analysis of the images presented in the previous point. First, the LSPIV methodology was assessed using the images with fluorescent particles and UV illumination. That methodology requires pre-processing of images through a sliding background (SLB), which eliminates the background of the images and particles that remain still between frames. Then the SSIV and BIV techniques were used to obtain velocity fields from the unseeded and LED-illuminated experiments. SSIV uses the same SLB image pre-processing to remove the background from the analysis and satisfactorily trace the movement of air bubbles and surface water reflections generated by raindrops. Additionally, BIV implements a previous binarization of the grayscale images to highlight bubbles from a determined threshold. Finally, a slight variation of the LSPIV methodology named LSPIVb was implemented, adding the binarization pre-process also for seeded UV experiments. Therefore, the image velocimetry techniques differ in the pre-processing of the images and the experiments used for the analysis. A diagram of the procedure followed for each technique is presented in Fig. 3, which includes a common PIV cross-correlation analysis and a post-



processing of the velocity results. The different image velocimetry techniques and the steps of the analysis are further explained below.

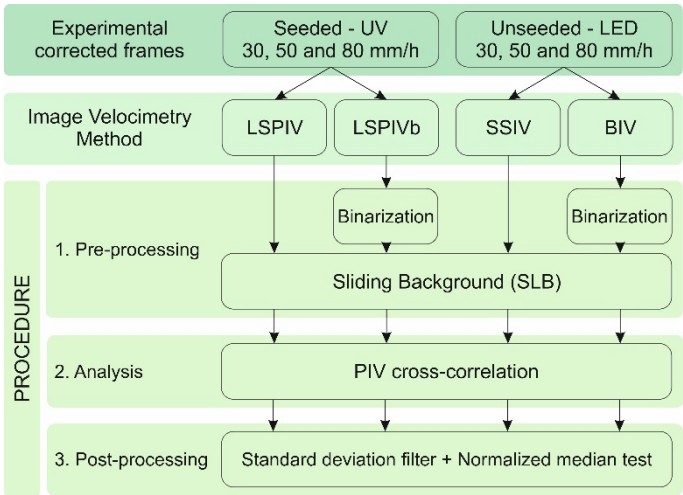


**Figure 3: Diagram of the procedure performed to obtain velocity fields from video frames for the different image velocimetry techniques.**

### 2.2.1 Pre-processing

As seen in Fig. 3, the first step in the analysis procedure was the pre-processing of the images depending on the technique
employed. The objective of this part of the analysis was to optimize the input images through different strategies to enhance the objective tracers and then measure their movement with PIV algorithms in the next step. The specific procedure followed for each technique is now described:

LSPIV velocity determinations were undertaken using the experimental videos with fluorescent tracers and UV illumination as in Naves et al. (2019a). The pre-processing in the implementation of this method consists of a sliding background filter.
This filter compares the gray values of the pixels of a frame with the same pixels of the following one, turning those pixels with a certain percentage of agreement between them to black. Therefore, it is necessary to determine a certain threshold of similitude to remove the background of the images and the particles that stay still between frames, whilst keeping those that are transported by flow. Thus, it is possible to avoid interferences in the cross-correlation of the particle movement that may reduce the mean velocity obtained.

In the case of the LSPIVb procedure, a binarization of the images was performed in the image pre-processing, prior to applying the sliding background filter. This converts the grayscale images to binary images, turning the pixels with gray values greater than a certain threshold to white and all other pixels to black. Due to this binarization, the subsequent sliding background filter is applied considering a threshold of 100% of similitude to remove the pixel, so only the binarization threshold should be adjusted. The procedure seeks to better remove background and shadows from the original frames in the estimation of velocities
from the seeded and UV illumination experiments.





In contrast to LSPIV and LSPIVb, the SSIV technique was applied to the images taken from the unseeded and LED illuminated experiments. SSIV is analogous to LSPIV, applying the sliding background filter below a certain threshold to remove the background of the original images. In this technique, the pre-processing of the images seeks to analyze the movement of both surface air bubbles and water reflections generated by raindrops. Lastly, the procedure of the BIV technique additionally

includes an image binarization filter, as was used in the LSPIVb procedure, to isolate air bubbles to be used as tracers in the analysis of the images taken from the unseeded experiments. The assessment of these methods, which obtain velocity fields directly from videos recorded from the catchment surface without adding particles, is an important task in order to study the feasibility of implementing this technology in real catchments, which would be an interesting source of velocity data to calibrate shallow water and dual drainage models.

### 160 2.2.2 PIV cross-correlation

The different alternatives of image processing, depending on the image velocimetry technique (LSPIV, LSPIVb, SSIV or BIV), were applied to 60 s of images taken in steady flow conditions, and the resulting frames were then analyzed by the PIV image software PIVLab (Thielicke and Stamhuis, 2014). This software performs a cross-correlation analysis between consecutive frames, which are divided into different interrogation areas (IA), to obtain the mean displacement vector for each

of the IA. The size of this IA is a parameter that must be adjusted as a function of the mean displacement in order to achieve suitable results. To compute the correlation matrix in the frequency domain, the Discrete Fourier transform (DFT) was proposed, calculated using a fast Fourier transform (FFT). Two passes of a multi-pass window deformation algorithm were used in the present work, halving the window size at the second pass to achieve a higher spatial resolution. The searching area (SA) matches with the IA and 50% of overlapping was selected in all cases in the present work.

### 170 2.2.3 Post-processing

Two filters for the detection of spurious vectors were applied to the velocity fields obtained. First, the results were filtered to remove those velocity vectors that differed four times the standard deviation from the mean velocity of the individual velocity fields. Then, the normalized median test was applied in a 3×3 neighborhood as proposed in Westerweel and Scarano (2005). The values of the two parameters of this filter were $\varepsilon = 0.15$ and the threshold = 3, which were investigated as optimum.

Outliers and missing data were removed and not replaced in any case. The average velocity field was obtained from the 1500 velocity results in steady conditions obtained for each case of study, which makes them comparable to the results achieved in Naves et al. (2019a) using the LSPIV technique. In this case, the velocities were compared as measured from the movement of tracers without applying velocity indexes to estimate depth-averaged velocities.

### 2.3 Comparative evaluation of image velocimetry methods

The present work seeks to perform a comparative assessment of different image velocimetry techniques in realistic but controlled laboratory conditions. The repetitiveness of the experiments allowed the evaluation of techniques that require





different experimental setups varying the seeding or illumination. In addition, the properties of the simulated rain (Naves et al., 2020c) include the presence of raindrops in the analysis as a novel scientific contribution.

Following the specific procedure explained in the previous section (Sect. 2.2) for each technique, the first step of the analysis
was the individual assessment of the robustness of the velocity results achieved by each technique. This assessment was carried out in a manner similar to Legout et al. (2012). The key parameters of the procedure were varied one-at-a-time within reasonable ranges to investigate their influence on the average velocity results. The parameters considered were: a) the pre-processing parameter, which corresponds with the sliding background or the binarization threshold depending on the image velocimetry technique; b) the IA initial size in the cross-correlation algorithm; and c) the frame acquisition rate (FAR) of the
experimental videos. The entire analysis was focused on four areas of the model surface in order to separately consider different types of flow that are developed in real catchments.

Moreover, a detailed comparison of the mean velocity fields achieved from one minute of steady conditions using each technique was performed considering each of the three rain intensities and each of the four surface areas. In this way, it was possible to study the differences observed and discuss the possibilities of imaging velocimetry techniques in different flow
conditions during rain events. In this analysis, the LSPIV method is used as the reference technique since it was already validated in Naves et al. (2019a). Finally, the transferability of the previous imaging velocimetry techniques to field studies is discussed considering the previous results and an additional convergence analysis, which assesses the uncertainties of measuring velocities in transient conditions. Further details of the studied areas, the ranges of the parameters and the implementation methodology are included in the following Sect. 2.4, 2.5, and 2.6.

**2.4 Areas considered for analysis**

The urban drainage physical model considered in this study enables typical flows such as those that are developed in real catchments to be reproduced. Figure 4 includes the previous velocity field results obtained in Naves et al. (2019a) for the rain intensity of 50 mm h[-1] and the drainage basin of gully pot 2. As can be observed, the runoff generated by the rainfall simulator produces two perpendicular flows: a very shallow flow towards the curb, and a longitudinal curb flow with depths up to 10
mm that drains the runoff into the gully pots. In addition, some preferential drainage channels, where velocities are significantly increased, have been detected in the overland flow perpendicular to the curb due to irregularities in the model surface.

The comparative analysis performed in this study was focused on four specific areas of the model surface in order to assess the performance of the techniques considered for different types of water flows that may be found in real catchments. These are: straight one-main-direction overland flow as presented in Area 1, where a preferential channel with high velocities is
distinguished; a parallel overland flow out of the main drainage channels with very low velocities and water depths (Area 2); the curb flow gathering different secondary flows with a vertical boundary (Area 3); and a combination of the previous type of flows in the vicinity of gully pot 2 (Area 4). Figure 4 includes the specific position in the model surface of each area considered in the analysis performed.





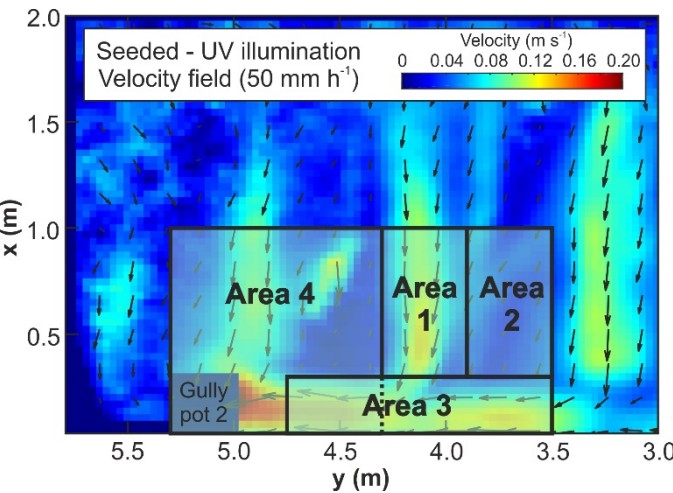

**Figure 4: Areas of the model surface analyzed with the image velocimetry methods considering different types of flows that are developed in urban catchments such as perpendicular drainage to the curb (Areas 1 and 2), curb flow (Area 3) and vicinities of gully pot 2 (Area 4). The velocity field plotted to understand the choice of the analyzed areas was taken from Naves et al. (2019a), where the LSPIV was satisfactory applied.**

## 2.5 Parameters ranges

As stated in Sect. 2.3, the first part of the analysis performed seeks to assess the robustness of the velocity results and their sensitivity to changes in the different input parameters of the procedure, depending on the image velocimetry technique used. The base parameters established in the analysis and their ranges of variation are given in Table 1. First, the FAR of the videos taken from the experiments was 25 Hz, which was established as the reference value since it is in consonance with most imaging devices available on the market. However, those cases in which only one of each two images (12.5 Hz) and one of each three images (6.25 Hz) are used for the analysis were also investigated in this study. This was to approach the conditions of worse devices that could already be installed, in accordance with the ideas developed in Leitão et al. (2018), where the use of surveillance cameras to obtain overland velocities was proposed.

**Table 1. Ranges of the parameters considered in the analysis of the different image velocimetry methods. Values in parentheses specify the base value of the parameters used as reference.**

| Image velocimetry method | FAR (Hz) | Pre-processing thresholds | | IA size (px) |
| --- | --- | --- | --- | --- |
| | | Binarization | SLB (%) | |
| LSPIV | 6.25 - 25 (25) | - | 0 - 50 (25) | 16 - 48 (32) |
| LSPIVb | 6.25 - 25 (25) | 0.15 - 0.35 (0.25) | 100 | 16 - 48 (32) |
| SSIV | 6.25 - 25 (25) | - | 5 - 25 (15) | 16 - 48 (32) |
| BIV | 6.25 - 25 (25) | 0.50 - 0.70 (0.60) | 100 | 16 - 48 (32) |

The variations produced in the velocities because of changes in the pre-processing parameters were also investigated. In the cases of LSPIV and SSIV, only the sliding background threshold was considered since binarization was not performed. In





contrast, LSPIVb and BIV required the definition of the binarization threshold, but the sliding background threshold had to be fixed at 100 % to delete those pixels that appear in white in two consecutive and binarized frames. The reference value of these

thresholds and their range of variation during the analysis were determined based on expertise and preliminary tests, resulting in variations of the SLB threshold from 0 % to 50 % for LSPIV and from 5 % to 25 % for SSIV; and binarization thresholds from 0.15 to 0.35 for LSPIVb and from 0.50 to 0.70 for BIV. Reference values were thus established as the mean value for each range.

The reference value for the IA size during the cross-correlation process was set following the recommendations in Raffel et al.

(2007) and Adrian et al. (2011). As the maximum velocity vectors are around 10 pixels/frame in absolute values, the reference interrogation area (IA) is set at 32x32 pixels. This assures the rule of thumb that displacements cover around 25 % of the total size of the IA. The range of IA sizes was established within 16 and 48 pixels.

**2.6 Implementation**

The variations of the velocity results because of changes in the parameters were analyzed by varying the reference value of

one determined parameter within its established range. In addition to the reference value, four different values were also considered using uniform steps for the IA size and the pre-processing parameter. For example, IA sizes of 16, 24, 32, 40, and 48 pixels were investigated whilst keeping the remaining parameters constant. As commented in Sect. 2.5, the FAR has been modified between 25, 12.5, and 6.25 Hz. This resulted in 11 different parameter sets for each of the four different image techniques, three different rainfalls and four different areas. Therefore, a total of 528 cases were considered when analyzing

the 1500 raw frames recorded in the seeded or unseeded experiments. The representation of velocity fields was performed through the Matlab toolbox 'pivmat' (Moisy, 2017).

**3 Results and discussion**

In the present section, the differences obtained in the velocity fields resulting from analyzing the experimental videos by four image velocimetry techniques (LSPIV, LSPIVb, SSIV and BIV) are presented and discussed. First, the sensitivity of the

different methods to image processing variables is investigated to assess their robustness and performance in analyzing shallow flows with the presence of raindrops. Then, velocity results are compared and the feasibility of using these techniques in transient flow conditions is analyzed.

**3.1 Sensitivity to image processing analysis**

The changes in the mean velocities obtained by varying the key parameters of the analysis for the three study areas, the four

imaging velocimetry techniques and the rain intensity of 50 mm h$^{-1}$ are shown in Fig. 5. The graph includes velocity results as the average of the mean velocities resulting from each pair of frames analyzed (1500 in the case of FAR of 25 Hz), plotting their standard deviation using whiskers. The reference value and the range of variation of the parameters considered, which





are the pre-processing parameter (binary threshold or sliding background depending on the technique used), the interrogation area size (IA), and the frequency acquisition rate (FAR), have been previously defined in Sect. 2.5. This analysis focused on

the intermediate rain intensity experiments, although similar results were obtained for 30 and 80 mm h$^{-1}$, which are included in the supplementary information. Generally speaking, Fig. 5 shows how the variation of the parameters within the established ranges did not produce significant variations in the mean velocity results. Therefore, the methodology and the imaging velocimetry techniques analyzed in this work are presented as being robust and the reference values can be considered when comparing the velocity results obtained by each technique.


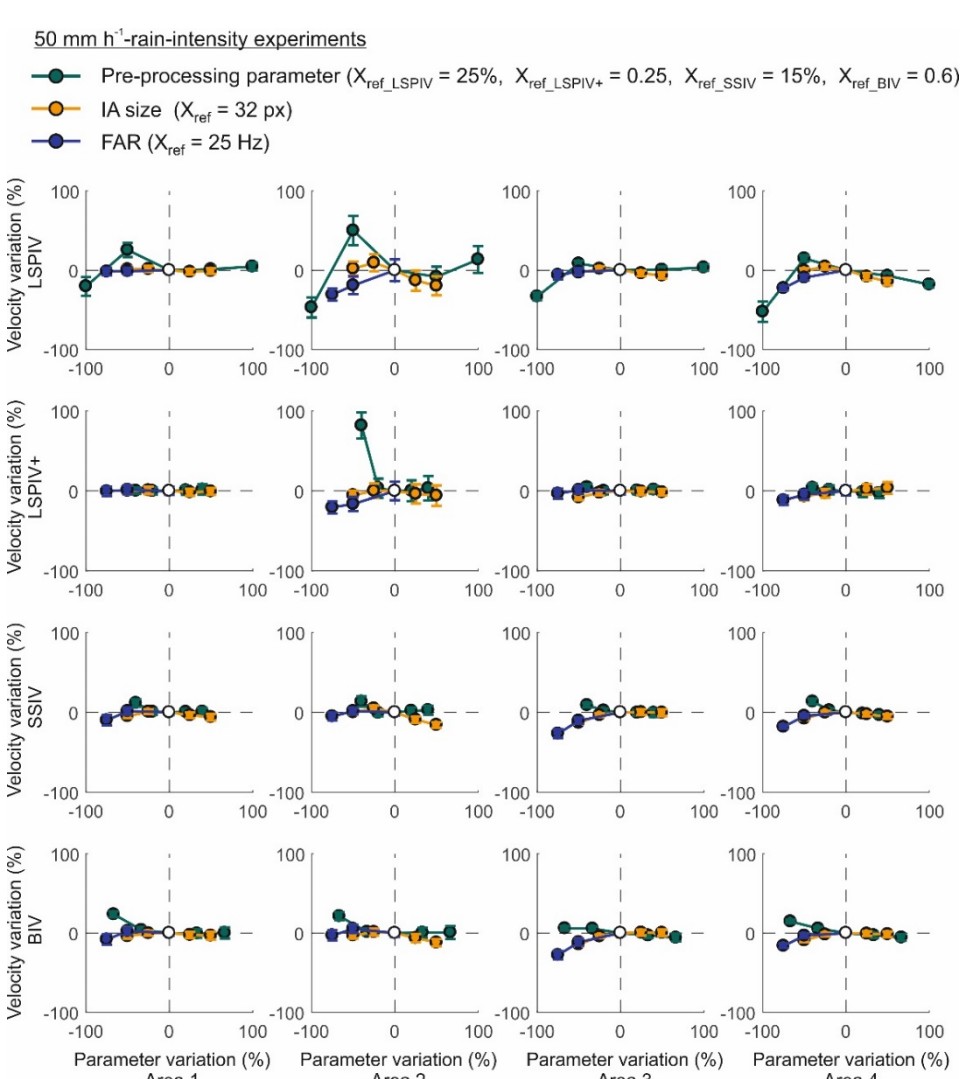

**Figure 5: Percentage of variation in the mean velocities when varying parameters of the analysis for the four studied areas (columns) and the four imaging velocimetry techniques considered (rows) in the case of 50 mm h$^{-1}$ rainfall. Mean velocity variability for the different pairs of frames analyzed are included using whiskers.**





Considering techniques that use seeded experiments (LSPIV and LSPIVb), the pre-processing parameter showed the greatest
influence on the results. This is due to the importance of removing particles that remain still on the model surface in order to
achieve reliable results. As seen in the results, the binarization considered in LSPIVb reduces this sensitivity to the pre-
processing parameter, except in Area 2 where the extremely shallow flow considered greatly favors particle deposition. The
type of flow developed in this area also increases the variability of the mean velocities depending on the pair of frames
analyzed, which remains low for the rest of the cases, as can be seen from the plotted whiskers. The techniques that analyze
the videos without particles (SSIV and BIV) are slightly less sensitive to variations in the parameters. In summary, the results
obtained by the imaging velocimetry techniques are presented as being quite stable and the velocities do not depend on the
parameters being within reasonable ranges established by expertise. Therefore, the velocity results obtained from the reference
parameter values are representative of each technique and can be used to assess their performance.

## 285 3.2 Velocity results comparison

The performance of each technique was assessed by comparing the velocity field results obtained in the study areas by each
technique for the three rain intensities considered. To do this and following the previous results (Sect. 3.1), the velocities
obtained using the reference values of the parameters (Table 1) have been considered for this comparison. In addition, the
LSPIV technique was used as the reference results since it had been previously validated in Naves et al. (2019a). The
comparative results are presented separately below for each study area in order to analyze the performance of each technique
in detail for the different types of flow developed on the model surface.

First, Fig. 6 shows the comparison of the velocity fields in Area 1, which corresponds to a main drainage channel perpendicular
to the curb where high velocities are developed. The velocity fields obtained and a derived disparity plot using the LSPIV
results as the reference were included in Fig. 6 for the three different rain intensities. The first result to highlight is that the
implementation of binarization in the pre-processing of the frames introduced no significant improvements in the velocity field
results. In contrast, very important differences were observed by comparing techniques that analyze videos with (LSPIV and
LSPIVb) and without (SSIV and BIV) particles. Considering the results for the lowest rain intensity (first row), all the
techniques presented a similar velocity distribution, although a gap of approximately 0.05 m s$^{-1}$ was obtained for the unseeded
techniques. This gap is because different types of tracers are being analyzed in each case and, while fluorescent particles are
transported in suspension inside the flow, bubbles and water reflections follow the higher velocities developed on the water
surface. Therefore, all the techniques obtained very good performance for 30 mm h$^{-1}$ rainfall and it is deduced that lower
velocity indexes are required in the case of the unseeded techniques to convert the results to depth-averaged velocities, as
observed in previous references (Leitão et al., 2018; Martins et al., 2018; Naves et al., 2019a). However, the velocity fields
obtained for rain intensities of 50 and 80 mm h$^{-1}$ showed that both the SSIV as well as the BIV techniques presented important
differences regarding velocity distributions. In the case of the rain intensity of 50 mm h$^{-1}$, there is a reduction in velocities for
the unseeded techniques, which is greatly incremented as the velocities are roughly higher 0.10 m s$^{-1}$ for the LSPIV technique.
Considering the 80 mm h$^{-1}$ results, it can be deduced that this unexpected decrease in velocities is clearly related with the rain





intensity, since the perturbations in the velocity results started to occur for lower LSPIV velocities and to a greater extent as the rain intensity increases.

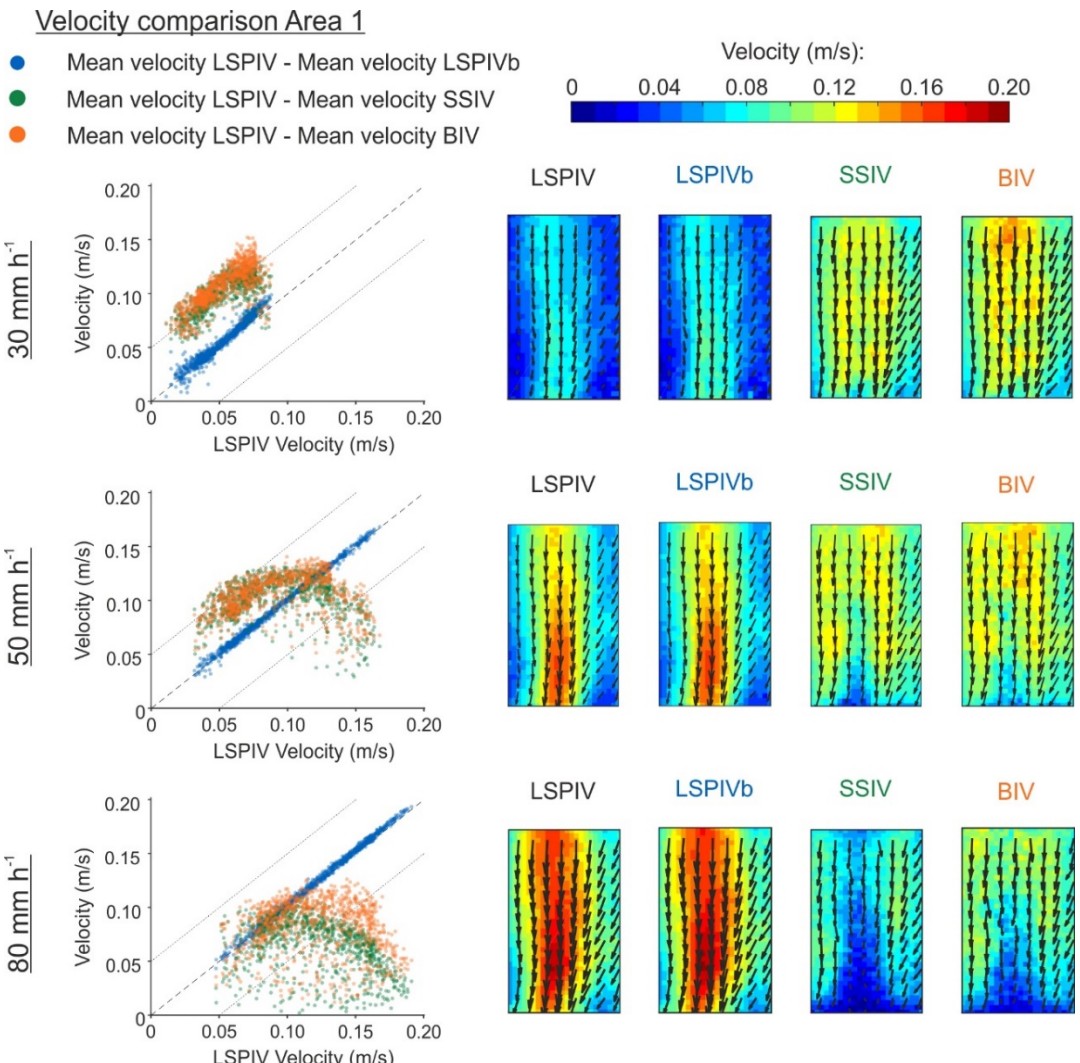

**Figure 6: Velocity comparison between imaging velocimetry techniques results (LSPIV, LSPIVb, SSIV and BIV) in Area 1 for the three different rain intensities (30, 50, and 80 mm h$^{-1}$). The velocity fields obtained for each case are also plotted for a qualitative comparison.**

This phenomenon can be explained by the combination of two different processes. First, the reduction in the gap between velocities obtained from the seeded and unseeded techniques for LSPIV velocities lower than 0.10 m s$^{-1}$ is explained because of the turbulences generated by the raindrop impacts, which decrease the flow surface velocity. The velocity index to estimate depth average velocities therefore depends on the rain intensity. Then, the inability of the unseeded techniques (SSIV and BIV) to measure the highest velocities is not produced because of a lack of tracers since, as can be observed in the videos provided in Naves et al. (2019b), the number of bubbles in that area increases with the rain intensity. The problem is caused by the





erratic trajectory of the bubbles observed in the unseeded videos for higher rain intensities, also due to the impact of raindrops on the water surface. It is assumed that raindrops, when falling, interrupt part of the existing flow and produce acceleration of the flow in all directions in the surrounding area to the impact (Kilinc and Richardson, 1973). The non-artificial bubbles used as tracers for the SSIV and BIV techniques are highly affected by these accelerations, producing very fast, random and major changes in the position of the tracers; these are exacerbated as the velocity of the bubbles and the rain intensity increases. This

avoids cross-correlation algorithms obtaining displacements of tracers and results in erroneous velocities. The mass of the fluorescent particles used as tracers in the case of LSPIV and LSPIVb, with a density slightly higher than water, confers themselves inertia to avoid such sudden movements and allows cross-correlation between consecutive frames.

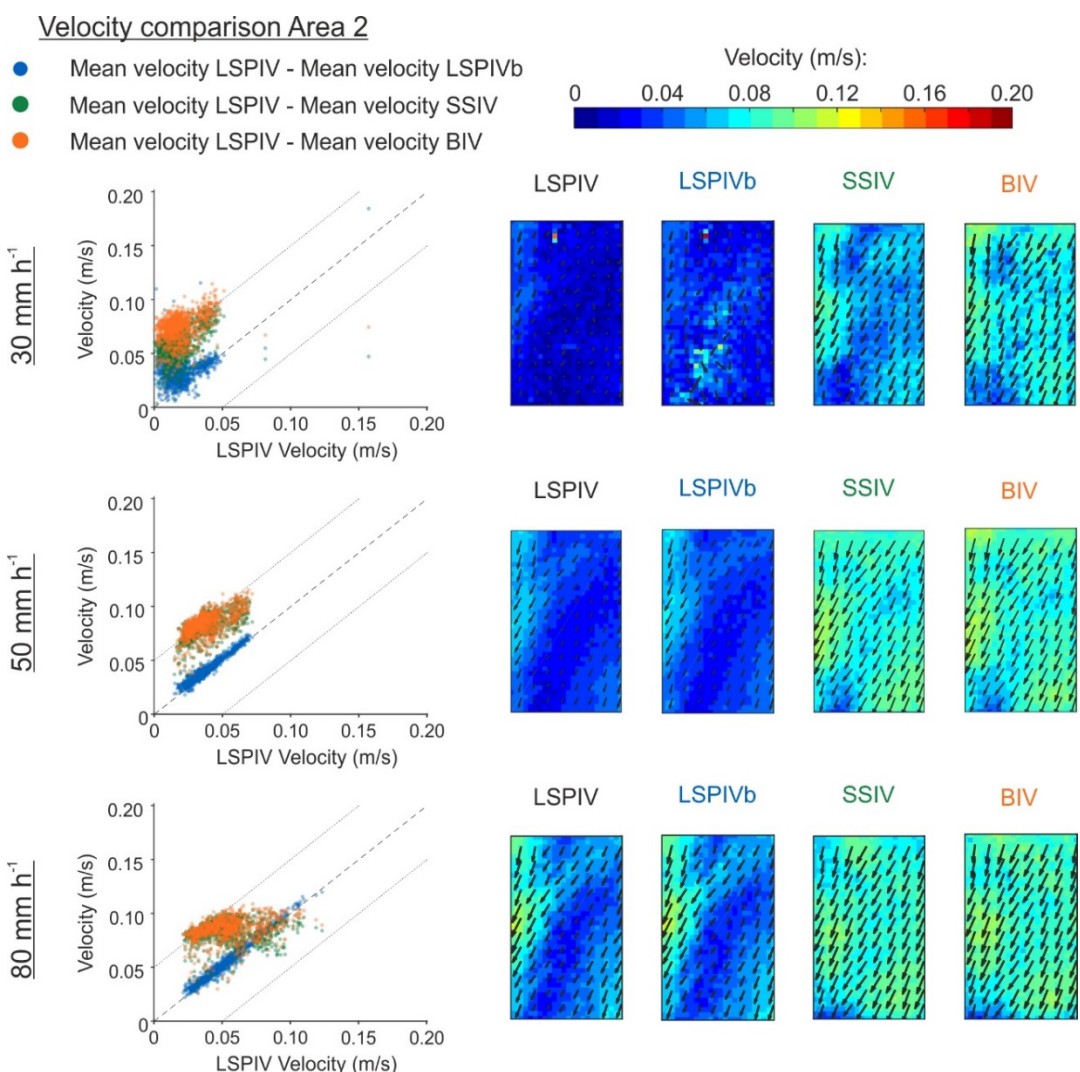

**Figure 7: Velocity comparison between imaging velocimetry techniques results (LSPIV, LSPIVb, SSIV and BIV) in Area 2 for the**
**three different rain intensities (30, 50, and 80 mm h⁻¹). The velocity fields obtained for each case are also plotted for a qualitative comparison.**





Despite these problems in measuring high velocities with high rain intensities, the use of bubbles as tracers can be an opportunity to measure velocities in extremely shallow flows where the particles tend to be deposited, as can be seen on the sides of the drainage channel in the velocity fields of Fig. 6, especially for lower rain intensities. This is better assessed in the

analysis of the results in Area 2 (Fig. 7), which is attached to Area 1 and considers a perpendicular flow to the curb, out of the main drainage channels and where very shallow flows are developed.

As can be seen in Fig. 7, there is a central zone in Area 2 where the methods that use particles as tracers (LSPIV and LSPIVb) are not able to obtain velocity data (dark blue areas in the respective velocity fields). Despite this zone, which corresponds with the lowest water depths, becoming smaller as the rain intensity is higher and the water depths increase, it is necessary to

use the SSIV or BIV techniques to obtain reliable results in the whole study area. As can be checked in the recorded videos, this is because bubbles are able to pass through the extremely shallow areas where fluorescent particles tend to be deposited. Regarding the influence of the rain intensity on the results, a similar behavior to that for Area 1 is observed but in this case, only the experiment with the highest rain intensities seems to be affected, due to the low velocities registered.

Figure 8 presents the comparison between the velocity results in Area 3. The velocity fields in this area refer to the channel

attached to the curb that runs perpendicular to the flows presented in Areas 1 and 2, and where most of the overland flows are gathered on their way to the drain inlet. Figure 8 only includes velocity fields for the rain intensity of 50 mm h$^{-1}$ since no interesting differences were found between rain intensities; the rest of the velocity field can be consulted in the supplementary information. In the recorded videos of the unseeded experiments, it was observed that raindrop impacts do not produce bubbles in that area due to the greater water depths and moreover, existing bubbles cannot access to this flow from the rest of the

catchment and stay retained in the confluence of flows. This results in a lack of tracers and thus the impossibility of measuring velocities correctly with the SSIV and BIV techniques. In contrast, LSPIV and LSPIVb present a high density of particles and accurate velocity results with a high concordance between both techniques. Therefore, it can be concluded that it is not possible to measure velocities in those conditions without the presence of bubbles using the SSIV technique to trace water reflections, so the use of particles as tracers is highly recommendable in this type of complex flows.




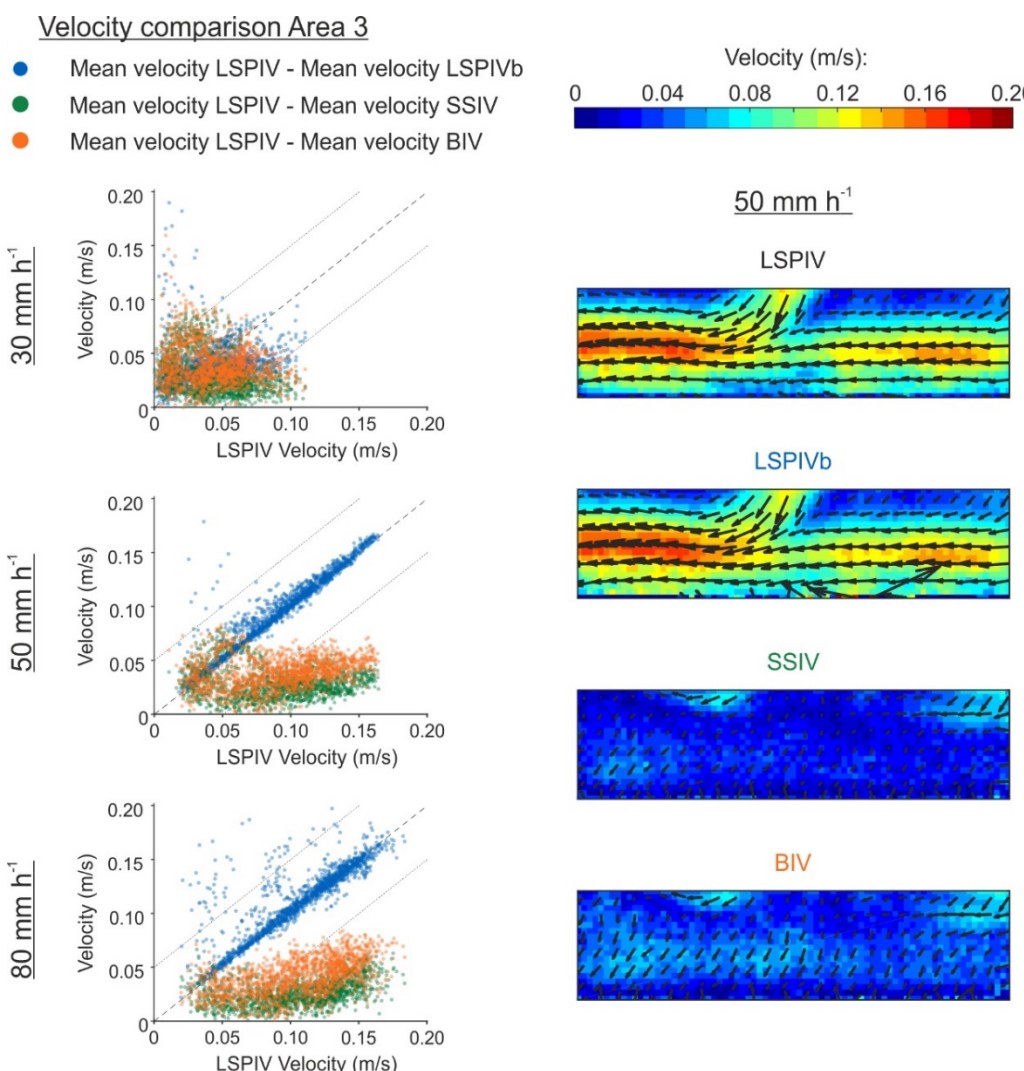

**Figure 8: Velocity comparison between imaging velocimetry techniques results (LSPIV, LSPIVb, SSIV and BIV) in Area 3 for the three different rain intensities (30, 50, and 80 mm h⁻¹). The velocity fields obtained for the case of 50 mm h⁻¹ are also plotted for a qualitative comparison.**

Finally, Fig. 9 shows the velocity fields obtained in Area 4 for the case of 50 mm h⁻¹ and the comparison between the results obtained by using the four imaging velocimetry techniques and the three rain intensities. The velocity fields for the rain of 30 and 80 mm h⁻¹ can be consulted in the supplementary information. This area covers the vicinity of gully pot 2 and is a combination of the previous cases studied considering overland flows perpendicular to the curb, such as those in Areas 1 and 2, and the curb flow analyzed in Area 3. The results plotted in Fig. 9 agree with the observations made for the previous areas and confirm the insights achieved.

Figure 9: Velocity comparison between imaging velocimetry techniques results (LSPIV, LSPIVb, SSIV and BIV) in Area 4 for the three different rain intensities (30, 50, and 80 mm h$^{-1}$). The velocity fields obtained for the case of 50 mm h$^{-1}$ are also plotted for a qualitative comparison.

As seen in Naves et al. (2019a), LSPIV obtained suitable results for the entire model surface, with some difficulties in extremely shallow flows with depths of around 1 mm where particles tend to be deposited. Then, the velocity fields showed a good correspondence between LSPIV and LSPIVb and between SSIV and BIV with slightly different performance of methods using binarization pre-processing (LSPIVb and BIV). Binarization removes pixels that seem to be still between consecutive frames more frequently since it does not consider the different shades of gray. Although this may improve velocity results in

very shallow areas because all the deposited particles and still tracers are removed, binarization may also remove tracers in





motion that are overlapped in consecutive frames, obtaining noisier velocity results (see velocity fields for 30 and 80 mm h$^{-1}$ in the supplementary information). In the case of the lowest rain intensity, all the techniques resulted in similar velocity distributions with a gap between the seeded and unseeded techniques of around 0.05 m s$^{-1}$, which is due to the different tracers analyzed. The use of bubbles and water reflections as tracers results in higher velocities in the case of SSIV and BIV because

they are transported on the water surface in contrast with the in-flow transportation of particles. This gives techniques that use bubbles as tracers a great opportunity to measure velocities in extremely shallow flows where particles tend to be deposited. However, SSIV and BIV are more affected by the impact of raindrops unexpectedly reducing their velocities as the rain intensity is increased, especially for high velocity flows. These techniques also presented problems in the flow attached to the curb because of the absence of bubbles in that area.

**3.3 Transferability to field applications**

The assessment of different imaging velocimetry techniques and the analysis of the influence of different factors on the velocity results contribute to understanding how these methodologies could be adequately transferred to real urban catchments. Although the reliability of the velocity data obtained using the unseeded techniques (SSIV and BIV) is compromised for high rain intensities, previous results showed unseeded techniques as a very promising solution for measuring velocities in field

applications, even in extremely shallow flow conditions. This is because of the good results obtained for low rain intensities and the benefits in terms of simplicity in their implementation by not having to add artificial particles to the flow. The use of these techniques in urban catchments would favor new velocity data sources to calibrate physically-based urban drainage models, such as surveillance cameras (Leitão et al., 2018) or even unmanned aerial vehicles, which have already been used in river flow measurements (Lewis and Rhoads, 2018; Pearce et al., 2020). At the same time, the use of particles in the LSPIV

technique enabled velocities in complex flows such as those developed in Area 3 of the present study (Fig. 8) to be measured, and the velocity results obtained were suitable even for high rain intensities, since particles are less affected by raindrop impacts. Therefore, the use of these seeded techniques may also be very interesting in real urban catchments in order to analyze specific areas of special relevance, such as for example the vicinity of manhole inlet grates, where more complex methodologies could be implemented.

Besides the ability of each technique to adequately measure overland flow velocities in urban catchments with shallow water flows and the presence of raindrops, it is important to assess the minimum requirements of the recording devices when evaluating the feasibility of using these techniques in field studies. This work has proven that a low frequency acquisition rate of 6.25 Hz could be used without incorporating significant errors into the velocity results obtained. However, in light of the problems observed in Leitão et al. (2018) for rates lower than 20 Hz using only water reflections as tracers, this requirement

varies according to the magnitude of the measured velocities and the tracer used. Therefore, it is deduced that artificial particles and naturally generated bubbles considered in the present study favor cross-correlation when the time step between frames is increased. The outdoor study carried out by Leitão et al. (2018) also concluded that SSIV provides robust results analyzing images with a resolution as low as 256×144 pixels for an area of around 5 m². These values are easily overcome by most



imaging devices available on the market so, although low acquisition rates and image resolutions will decrease the precision

and quality of the velocity results, these are not presented as a major constraint to transferring imaging velocity techniques to

real applications in urban catchments.

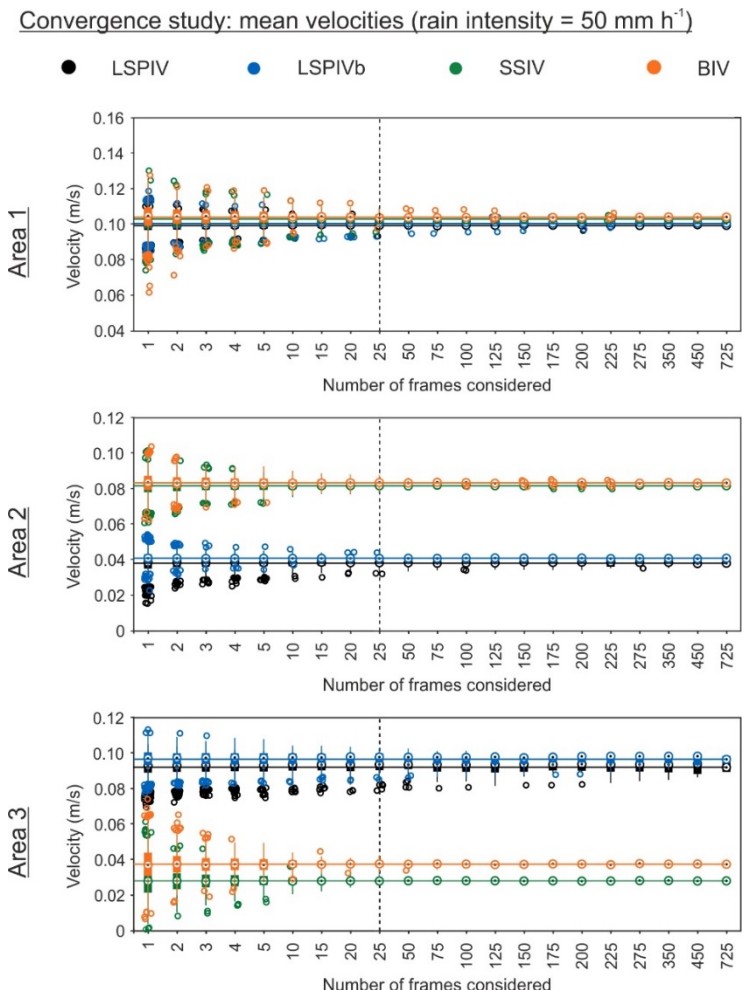

**Figure 20: Mean velocity convergence study for rain intensity of 50 mm h$^{-1}$ and Areas 1, 2, and 3. The horizontal line represents the**
**mean velocity considering all the frames available (1500) for LSPIV (black), LSPIVb (blue), SSIV (green) and BIV (orange)**
**techniques. Then, the variability in the mean velocity when the frames are divided into groups of different numbers of frames was**
**represented by boxplots.**

Finally, the analysis presented in this article has been performed for steady flows comparing average velocity fields obtained

from one minute of steady conditions. This allows the comparability of the results with those obtained in Naves et al. (2019a)

and reduces the uncertainties in velocity estimations, but is not representative of field conditions. Therefore, another important

point to address is the uncertainties assumed when using these techniques in transient flows, such as those that would be

recorded in real catchments. A convergence analysis is presented in Fig. 10 for the experiments with 50 mm h$^{-1}$ of rain intensity





and for Areas 1, 2, and 3, showing the variations in the mean velocities that arise when the number of frames considered is reduced. The first result to highlight is that the mean velocities obtained from each method and from each area follow the insights presented and discussed in Sect. 3.2. In addition, the variations in the mean velocity remain low considering 25 frames in the analysis, which corresponds to one second in the present case study. This value is considered to be enough to ensure reliable velocity results analyzing transient flows and to enhance the possible usability of these imaging velocimetry techniques in future real field studies, meriting further investigation. Convergence results for other rain intensities and for Area 4 are similar and can be consulted in the supplementary information.

## 4 Conclusions

In this study, the performance of different seeded and unseeded imaging velocimetry techniques has been assessed from videos of the overland flow generated by three different rain intensities in an urban drainage physical model. These techniques use artificial-seeded particles or existing bubbles and water reflections as tracers to estimate surface velocity distributions. The influence of the rain intensity on the reliability of the results has also been explored as a novel scientific contribution by comparing the velocity fields achieved from each study case. Based on the results obtained, the following conclusions can be drawn:

- The variation in the parameters of the imaging velocimetry techniques within the established ranges did not produce significant variations in mean velocity results. Therefore, the methodology and the imaging velocimetry techniques analyzed in this work are presented as being robust, and expertise can be used to set the required parameters for the analysis.

- Both seeded and unseeded techniques provide suitable velocity distributions for lower rain intensities, observing a gap of approximately 0.05 m s$^{-1}$ in velocities due to the different tracers being analyzed. The mobilization of bubbles on the water surface results in higher velocities for unseeded techniques, but gives them a great opportunity to measure extremely shallow flows where particles tend to be deposited, and to be applied in field applications where the distribution of artificial particles in urban catchments during rain events is not trivial.

- Unseeded techniques are highly affected by raindrop impacts. First, the gap between seeded and unseeded techniques is reduced as the rain intensity is increased, so rain intensity should be considered to determine the velocity index for estimating depth-average velocities. Then, raindrop impacts also produce fast and random changes of position of the bubbles used as tracers, leading to erroneous velocities for the highest rain intensities.

- The rapid convergence of velocity results makes the analysis of transient flows feasible. This fact, as well as the not very demanding requirements of the recorded videos, favor the transferability of these techniques to field studies, where they can be used as a novel tool to obtain runoff velocities in order to calibrate physically-based urban drainage models.





Thus, this work highlights not only the feasibility of using both seeded and unseeded imaging techniques to obtain surface
velocities in shallow flow conditions and with the presence of raindrops, but also the importance of considering rain properties
to interpret and assess the results obtained by these type of techniques. Future research should be oriented towards the
application of these techniques in real urban catchments.

**Author contribution**

J.N. performed the experiments with the supervision of J.A., J.P. and J.S.. J.N. processed the experimental data with the
supervision of J.G. and J.A. J.N., J.G. and J.A. conceptualized the study, interpreted the results, structured the paper and
prepared the original draft. All authors revised the manuscript critically for important intellectual content.

**Competing interests**

The authors declare that they have no conflict of interest.

**Acknowledgments**

The project receives funding from the Spanish Ministry of Science and Innovation under POREDRAIN project RTI2018-
094217-B-C33 (MINECO/FEDER-EU).

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
