# Peer review of "Assessing different imaging velocimetry techniques to measure shallow runoff velocities during rain events using an urban drainage physical model"

_Hydrology and Earth System Sciences, 2020_

## Referee Comment (RC1) · Anonymous Referee #1 · 8 Jun 2020

General comments

This study assesses four image based velocimetry techniques for measuring water velocity in shallow flows as would be observed in overland flows over paved surfaces during rainfall events. The problem is physically challenging, and the authors use a dedicated lab setup to assess these techniques with an eye to evaluating their potential for more difficult and varied conditions in the field. The work is derived from a larger project that has led to a number of significant publications over the last couple of years and is a direct extension of a 2019 paper that validated one of the approaches used

(LSPIV). The validated approach is used as the reference condition for the current paper. The paper overall was well written and the methods appeared to be suitable for assessing the other velocimetry techniques.

Despite the quality of the work, the authors in my opinion are too positive about the results. In looking at the results from my reviewer's perspective, it appears that the unseeded techniques are not suitable for measuring velocities in shallow flows. Even in relatively straight flows with low precipitation, there is an offset between the unseeded techniques and the LSPIV results that is not well explained. It is not clear to me how the magnitude of this offset could be predicted without controlled tests. As the precipitation intensity increases, the error in the unseeded techniques increases to the point where the results are no longer even correlated with the validated technique. In these conditions I would argue that the unseeded techniques are simply not suitable. Despite this, many of the statements in the discussion and conclusions are quite positive about the techniques. The optimism seems to be related to other studies or results that are not included in the current paper. Something needs to be adjusted, either by including those results (maybe cases without any precipitation at all?) or by drawing sharper lines about which techniques are reliable in different conditions.

Questions/Comments:

148 – Is the LSPIVb procedure significantly different than the LSPIV? In reading the methods I thought that the results might arrive at the same point as each requires a threshold, one applied to the difference, the other to the base images and then the difference is then calculated. The results also show that they are nearly the same. The point of the LSPIVb analysis is not emphasized in the paper. What is the motivation for evaluating this technique? It is not really discussed in the results or appear in the conclusions to a significant extent. Does it 'better remove background and shadows. . ."? Should other people use it instead of the regular LSPIV?

Minor issues

16 - complex sentence. should split into one about the natural tracers and the second about the raindrop impacts.

24 – replace 'Specifically' with 'However'?

47 – sentence starting with 'For instance' is not clear to me. Should be rewritten in a more direct style.

61 – sentence starting with 'That study' is too complex. Should be split into two ideas.

87 and 106 – is Naves et al 2019b an archive? Data availability should be clarified.

101 - Best to say what was done step by step. e.g. Videos were recorded at 4K resolution and 25Hz. 1500 frames (eq. to 60 s) were extracted from the longer recording for analysis.

117 – 'estimate' is better than 'obtain' for this sentence.

117 – 'from the analysis of the images presented in the previous point' is not necessary.

120 – so all particles are assumed to be moving? Is this realistic? Is there a velocity threshold?

156 – restatement of the aim/objective. Not necessary in the methods.

166 – Description of correlation matrix calculation is too brief. Need to help readers who may want to apply this technique themselves. Is this following what was done for other publications?

174 – not clear what you mean by 'which were investigated as an optimum'.

180 – again repetition of aim, but shouldn't be necessary.

183 – novelty should be addressed in intro with aim and objectives.

193 – more repetition of the aim

195 – I think that the reference technique statement should also be used as a scop-

ing statement at the end of the introduction with the aim/objectives. Mixing it in here reduces the clarity of what is being done and what the starting point for the new contribution is.

224 – suggest 'typical of' rather than 'in consonance with'

226 – 'This was to approach the conditions of worse devices . . .' is not clear.

261 – Acronym not introduced until next page (FAR)

279 – what type of flow specifically is in the area? The shallowness? Should be clarified.

284 – should note that there is a degradation of quality with FAR, as expected.

317 - change phrasing. The current sentence uses a double negative. i think you mean that the unseeded technique is not able to measure the highest velocities, but i'm not sure.

322 – 'non-artificial' is again kind of a double negative. Just say what it is - the natural bubbles.

325 – suggest 'prevents' rather than 'avoids'.

325 – 'from' obtaining

414 – should be Figure 10

445 – do you mean that the problem is not trivial?

---

## Referee Comment (RC2) · Rolf Hut (Referee) · 7 Jul 2020

Naves at all present a lab comparison of different particle imaging velocimetry techniques under (indoor) rainy conditions. I believe that analyzing this is a valuable addition to the scientific literature. I especially am happy that the authors have shown that not all PIV techniques perform equally under different rainy conditions, something very relevant when analyzing (urban) flood impact from video images.

I commend the authors on their thorough effort of making the data on which they build

their conclusions available to the public in true Open Science spirit.

I have a few minor issues with the paper in its current form, but am overall of the opinion that this paper should be published in HESS.

(standard) disclaimers for reviewing:

- my main line of research is not in urban flooding, but in larger scale hydrology and in designing observational techniques. I have reviewed the paper through this lens.

- I'm not an English native speaker, nor do I believe it is the job of scientists not trained as copywriters or editors to review each others placement of comma's and style forms used, I have therefore looked at the science presented, not at any language issues.

Minor issues:

- The paper focusses on the application of urban flooding of streets and this is reflected in the literature cited. In river hydrology there are quite some papers also looking into using seeding for better LSPIV results. Multiple papers by Flavia Tauro and her team come to mind. Perhaps (but I'm not sure) adding these in the introduction would better frame the current research.

- While the authors do make all their data available, and they do state which software packages they use for part of their analyses, it is impossible for me to check their results, since the code they use to generate their results is not shared. I would like to ask the authors to upload the code that generates the figures presented in the paper to Zenodo and cite it in the manuscript. This would also facilitate reproducing the result of this study, or expand on it.

- The authors make use of the "jet" colormap for their figures, a choice that is known to results in figures that highlight differences not present in the data. (See among others https://www.jstor.org/stable/24862699?seq=1) Please switch to a different colormap. (this is a pet peeve of mine)

Concluding: I really like the paper, the science, as presented, is sound although the actual claims cannot be verified without the software that generated their results shared alongside the paper.
* * *

---

## Author Comment (AC1) · 20 Jul 2020

**Response to reviewer 1**

**General comments**

This study assesses four image based velocimetry techniques for measuring water velocity in shallow flows as would be observed in overland flows over paved surfaces during rainfall events. The problem is physically challenging, and the authors use a dedicated lab setup to assess these techniques with an eye to evaluating their potential for more difficult and varied conditions in the field. The work is derived from a larger project that has led to a number of significant publications over the last couple of years and is a direct extension of a 2019 paper that validated one of the approaches used (LSPIV). The validated approach is used as the reference condition for the current paper. The paper overall was well written and the methods appeared to be suitable for assessing the other velocimetry techniques.

**Response:**

We appreciate the reviewer for the time invested in reviewing our manuscript and for the positive evaluation of our work. We are grateful for the detailed review and believe that the constructive comments and suggestions will lead to a deeper and clearer analysis of the results presented in this article, contributing to significantly improve the quality of the manuscript. In this document, we present the responses to the reviewer comments indicating how we will address them in the revised version of the manuscript.

Despite the quality of the work, the authors in my opinion are too positive about the results. In looking at the results from my reviewer's perspective, it appears that the unseeded techniques are not suitable for measuring velocities in shallow flows. Even in relatively straight flows with low precipitation, there is an offset between the unseeded techniques and the LSPIV results that is not well explained. It is not clear to me how the magnitude of this offset could be predicted without controlled tests. As the precipitation intensity increases, the error in the unseeded techniques increases to the point where the results are no longer even correlated with the validated technique. In these conditions I would argue that the unseeded techniques are simply not suitable. Despite this, many of the statements in the discussion and conclusions are quite positive about the techniques. The optimism seems to be related to other studies or results that are not included in the current paper. Something needs to be adjusted, either by including those results (maybe cases without any precipitation at all?) or by drawing sharper lines about which techniques are reliable in different conditions.

**Response:**

Based on the comment of the reviewer, we have reread the manuscript and we agree that the discussion and conclusions appear too positive considering results presented for unseeded methods (BIV and SSIV). This may lead journal readers to confusion if a clear and contextualized interpretation of the results is not included. The positiveness showed is due to the great potential of unseeded techniques as a tool to obtain runoff velocity data from media sources commonly available in urban environments, such as surveillance cameras, traffic cameras, or even social media. From our point of view, this is a novel and powerful data source with a great potential to

solve the current lack of surface runoff velocity data, which is key in the proper calibration of the increasingly more accurate 2D-1D dual urban drainage models that are currently being developed. The use of these data sources has been recently introduced in the field of urban drainage (Leitão et al. 2018, de Vitry et al. 2020). In addition, in contrast with the increasingly more common application of visualization techniques to rivers monitorization (Pearce et al. 2020, Tauro et al. 2016, Tauro et al. 2018), their use for urban runoff measurements is still limited to some initial applications on the analysis of the velocities in a stormwater storage facility (Zhu et al., 2019) and in simulated urban floods (Leitão et al. 2018), both without precipitation. Therefore, this is the first work where the influence of raindrop interference with the recorded images and their impact in the measurement of velocities is analyzed, and therefore is still much room for improvement in this novel implementation.

In this context, we consider as positive the results obtained from unseeded techniques for low rain intensities where the results correlate with LSPIV results, since two different techniques are being used. The different density and size of bubbles and artificial fluorescent particles explains the offset found for low rain intensities, since tracers are affected in different degrees by raindrop impacts and may be transported at different velocities. The gap obtained in this study between seeded and unseeded experiments may be thus interpreted as an indicator of the uncertainties that may appear when using visualization techniques in rainy conditions depending on the type of tracer. As commented by the reviewer, we think that the prediction of the magnitude of this offset is currently challenging, so further investigation on how rain impacts the transport of possible tracers appears as an interesting research line to reduce uncertainties in runoff velocity measurements. Finally, the different tracers analyzed also explain the different behavior of seeded and unseeded techniques when the rain intensity is increased. While raindrop impacts produce fast and random changes of position in bubbles, that are incremented with rain intensity, the higher density of fluorescent particles confers themselves inertia to avoid such sudden movements. We think that this great affection of raindrops in unseeded techniques, leading to erroneous results for high rain intensities, is an important result of the present work and we agree that this should have been clearer presented in the text.

We will thus revise the results and discussion chapters to clarify these points by, as suggested by the reviewer, drawing sharper lines between the performance of each methods, explaining deeply the offset obtained between seeded and unseeded techniques, and discussing the potential and possible challenges of visualization techniques to measure urban runoff velocities. Among other little modifications, the following sentence will be added in the Results and discussion and conclusions sections to clearly specify that unseeded methods are not working for high intensities:

*"However, the velocity fields obtained for rain intensities of 50 and 80 mm h$^{-1}$ showed that both the SSIV as well as the BIV techniques resulted in erroneous velocity distributions, being more affected the areas where greater velocities are developed"*

*"This also gives techniques that use bubbles as tracers an opportunity to measure velocities in extremely shallow flows where particles tend to be deposited. However, SSIV and BIV are more affected by the impact of raindrops leading to erroneous results for high rain intensities, especially for high velocity flows"*

*"Unseeded techniques are highly affected by raindrop impacts. First, the gap between seeded and unseeded techniques is reduced as the rain intensity is increased, so rain intensity would be considered to determine the velocity index for estimating depth-average velocities. Then, raindrop impacts also produce fast and random changes of position of the bubbles used as tracers, leading to erroneous velocities for high rain intensities. However, the ability of measuring extremely shallow flows where particles tend to be deposited, and their easy implementation without the need of adding artificial particles, make unseeded techniques worthy of future investigations as new source of runoff velocity data in urban catchments."*

The explanation of the offset between seeded and unseeded experiments will be completed in Section 3.2 and stated in conclusions as follows:

*"All visualization techniques presented a similar velocity distribution for the lowest rain intensity (first row), although an offset of approximately 0.05 m s-1 was obtained for the unseeded techniques. This offset is produced because the different tracers used in seeded and unseeded experiments, which are affected in different degrees by raindrop impacts and may be transported at different velocities. Considering the novel application of these techniques in presence of rain, it can be deduced that all techniques obtained a good performance for 30 mm h$^{-1}$ rainfall and that lower velocity indexes are required in the case of unseeded techniques to convert the results to depth-averaged velocities, as observed in previous references (Leitão et al., 2018; Martins et al., 2018; Naves et al., 2019a)"*

*"Both seeded and unseeded techniques provide suitable velocity distributions for lower rain intensities in case of unidirectional flows, observing an offset of approximately 0.05 m s-1 between them. This offset is a consequence of the different tracers used in seeded and unseeded experiments, which are affected in different degrees by raindrop impacts and may be transported at different velocities. Lower velocity indexes are thus required in the case of unseeded techniques to convert the results to depth-averaged velocities. In case of more complex flows, unseeded techniques are not able to adequately measure since bubbles have difficulties to follow the runoff generated."*

**Questions and comments:**

148 – Is the LSPIVb procedure significantly different than the LSPIV? In reading the methods I thought that the results might arrive at the same point as each requires a threshold, one applied to the difference, the other to the base images and then the

difference is then calculated. The results also show that they are nearly the same. The point of the LSPIVb analysis is not emphasized in the paper. What is the motivation for evaluating this technique? It is not really discussed in the results or appear in the conclusions to a significant extent. Does it 'better remove background and shadows. . ."? Should other people use it instead of the regular LSPIV?

**Response:**

This is an interesting comment of the reviewer. It is true that preprocessing of both LSPIV and LSPIVb depends on only one threshold, but the different purpose of these thresholds lead to different preprocessed images. First, the sliding background preprocess was applied in the LSPIV technique to remove all elements that remain fixed between two consecutive frames, including the road surface, other elements of the physical model, and immobile particles. The threshold used in this case corresponds a percentage of the grey value to consider that an element does not move since, although the pixel correspond to a immobile element, this value may slightly vary because of variations on water surface or raindrops interferences. In contrast, the binarization performed for LSPIVb technique seeks mainly to isolate the brightest pixels, which in this case will correspond with the fluorescence particles used as tracers. Then, we also remove deposited and immobile particles with the sliding background filter, but the binarization makes the previous margin of gray value used in LSPIV unnecessary.

Binarization is used in PIV studies (e.g. Zhou et al. 2013) to remove the remaining noise in raw PIV images, resulting in images where all the particles have the same intensity and thus have equal contribution to the correlation function. However, this preprocessing technique would lead to increase measurement uncertainty if the threshold value is not properly addressed (Raffel et al. 2018). The motivation of including binarization in seeded experiments (LSPIVb technique) was firstly consistency with the unseeded techniques found in the literature. While SSIV seeks to remove immobile features from frames through sliding background and analyze the movement of all other elements, BIV seeks to analyze only the movement of bubbles, which are identified as the brightest elements in the images, removing the rest of features from the images. When we applied these preprocessing procedures to seeded experiments images, we observed that the processed images were slightly different and the binarization (in LSPIVb technique) reduced the number of particles to analyze in the images. For example, this can be observed in the following figure (Figure R1), where the same frame obtained from seeded videos was preprocessed following the procedures for LSPIV and LSPIVb respectively.

[Figure]

Figure R1. Preprocessed frame for LSPIV and LSPIVb imaging techniques.

In view of the differences between the images and the use of binarization in the literature, we decided to include LSPIVb technique to investigate the influence of binarization in the analysis of seeded experiments. As noted by the reviewer, the achieved results with both seeded techniques are very similar with slightly higher velocities obtained by techniques that use binarization. This similarity indicates that, except particles and bubbles, cameras did not record many other moving elements that disturb the results, so binarization does not include significant benefits in these experiments. Another interesting difference observed is that techniques including binarization resulted in noisier velocity results. This may be due to the fact that, if the binarization is applied, the sliding background filter may remove parts of tracers in motion that are overlapped in consecutive frames since no different grey values are considered, which also might explain the slightly higher velocities obtained. This indicates that binarization, which may be useful to isolate tracers if seeded experiments are performed with natural or regular artificial illumination, should be used with care in future applications and it is not recommended if the non-binarized images results in good correlations.

A detailed explanation of the motivation of evaluating LSPIVb and a more extended discussion about the similar results obtained using LSPIV and LSPIVb will be added to Methods and Results sections for a better comprehension of the achieved results. Specifically, the following text will be added to the methodology (Sections 2.2 and 2.2.1 respectively) to clarify the motivation of LSPIVb:

*"Finally, a slight variation of the LSPIV methodology named LSPIVb was implemented to investigate the influence of binarization pre-process also in the analysis of seeded UV experiments. This strategy seeks to isolate the brightest pixels, which in this case correspond with the fluorescence particles, to ensure that other elements such as bubbles or water reflections are not interfering in the PIV analysis."*

> *"This filter ensures that only the fluorescent particles are being considering in the PIV cross-correlation, preventing possible small interferences that bubbles or water reflections may produce despite the special illumination."*

In addition, the similar results obtained by LSPIV and LSPIVb will be discussed in Results and discussion section as follows:

> *"Then, the velocity fields showed very similar results between LSPIV and LSPIVb and between SSIV and BIV with slightly higher velocities measured by methods using binarization pre-processing (LSPIVb and BIV). This similarity indicates that, except particles and bubbles, cameras did not record many other moving elements that disturb the results, so binarization does not includes significant benefits in these experiments. In addition, it has been observed that techniques that include binarization result in noisier velocity results (see velocity fields for 30 and 80 mm h-1 in the supplementary information). This may be due to the fact that, if the binarization is applied, the sliding background filter may remove parts of tracers in motion that are overlapped in consecutive frames since no different grey values are considered, which also might explain the slightly higher velocities obtained. Therefore, this filter should be used with care in future applications if it would be necessary to isolate tracers from other mobile elements."*

Finally, an additional point will be added to conclusions:

> *"The similarity found between LSPIV and LSPIVb and between SSIV and BIV indicates that binarization preprocessing has not significant benefits in these experiments since cameras did not record moving elements that significantly disturb the results. In addition, it has been found that this procedure lead to noisier results, so binarization should be used with care in future applications if it would be necessary to isolate tracers from other mobile elements."*

**Minor issues:**

16 - complex sentence. should split into one about the natural tracers and the second about the raindrop impacts.

**Response:**

The sentence will be split and slightly modified to simplify the text:

> *"First, the use of naturally-generated bubbles and water shadows and glares as tracers allows the unseeded techniques (SSIV and BIV) to measure extremely shallow flows. However, these techniques are more affected by raindrop impacts, which even lead to erroneous velocities in the case of high rain intensities."*

47 – sentence starting with 'For instance' is not clear to me. Should be rewritten in a more direct style.

**Response:**

The sentence will be rewritten in a more direct style as follows:

*"Zhu et al. (2019) achieved errors below 14% using this technique in a full-scale stormwater detention basin, although in some bordering points these could rise up to 44%."*

61 – sentence starting with 'That study' is too complex. Should be split into two ideas.

**Response:**

The sentence will be rewritten for a better understanding:

*"The presence of raindrops in the experiments can generate disturbances in the water surface and also interfere in the visualization of images, so that study used UV illumination and fluorescent particles as artificial tracers to satisfactorily address these issues."*

87 and 106 – is Naves et al 2019b an archive? Data availability should be clarified.

**Response:**

Yes, the reference is from a dataset published by the authors in the open access repository Zenodo. This includes videos, images and related information to replicate our study or produce new results. This will be specified within the text as follows:

*"The freely available experimental dataset (Naves et al. 2019b) described in Naves et al. (2020b) was used in this study for the assessment of different imaging velocimetry techniques.*

*"Examples of these images obtained from UV seeded and LED unseeded experiments, which are openly available for others to use in the dataset published by the authors (Naves et al., 2019b), are included in Fig. 2."*

101 - Best to say what was done step by step. e.g. Videos were recorded at 4K resolution and 25Hz. 1500 frames (eq. to 60 s) were extracted from the longer recording for analysis.

**Response:**

As recommended, the sentences will be rewritten to clarify the methodology:

*"During the experiments, videos were recorded at 4K resolution and 25 Hz. 1500 frames of steady flow (equivalent to 60 s) were then extracted from the longer recording and processed for analysis. To do this, frames were scaled and ortho-rectified using the known 2D coordinates of 28 and 24 reference surface points for each camera. Finally, the reference points placed in the intersection between the recorded areas of each camera were used to crop and join the images, resulting in raw images where 1 pixel corresponds to 1 mm in real-world coordinates."*

120 – so all particles are assumed to be moving? Is this realistic? Is there a velocity threshold?

**Response:**

Some of the particles used as tracers may settle on the road surface due to the extreme low depths developed in some areas of the road surface and to the rugosity of the concrete surface. While the rest of particles follow the runoff generated, these particles appear immobile in the images recorded. This can lead to erroneous velocity results when the PIV cross-correlation is performed, because the null velocity of these particles can reduce the mean velocity of the particles of a determined interrogation area. The sliding background preprocess avoids this issue removing the immobile particles and focusing the analysis on the particles that are being transported by the flow. In addition, this procedure does not produce any velocity threshold in the results since only the particles that remain immobile between two consecutive frames are removed from the analysis, as can be also checked in the previous work published by the authors (Naves et al. 2019a). However, as stated within the text, the seeded techniques are not able to measure velocities in areas with extreme shallow flows (Area 2) because the artificial particles cannot be transported, and unseeded techniques appear as suitable tools to be further explorer for these conditions.

The sentence will be rewritten to clearly explain the removal of immobile particles:

> *"That methodology requires pre-processing of images through a sliding background (SLB), which eliminates the background of the images and particles that remain immobile between frames. These particles, which are deposited due to the extreme low depths developed and the rugosity of the concrete surface, should be removed to avoid that the null velocities resulted from them condition the PIV analysis."*

156 – restatement of the aim/objective. Not necessary in the methods.

180 – again repetition of aim, but shouldn't be necessary.

183 – novelty should be addressed in intro with aim and objectives.

193 – more repetition of the aim

195 – I think that the reference technique statement should also be used as a scoping statement at the end of the introduction with the aim/objectives. Mixing it in here reduces the clarity of what is being done and what the starting point for the new contribution is.

**Response:**

We agree with the reviewer. The final statement of the introduction section will be completed to include clearer the aim, the novelties, and the starting point of the new contribution. This content will be removed from the rest of the manuscript where, as noted by the reviewer, it is not necessary.

> *"Therefore, experimental videos of the overland flow generated by three different rain intensities, under laboratory-controlled conditions and recorded with and without artificial particles, are used in this study to comparatively assess the performance of different seeded and unseeded imaging velocimetry techniques under rainy conditions. First, the sensitivity of the velocity results to the analysis parameters is investigated in order to test the robustness of each method. Then, the resulting velocity*

*fields are compared to analyze the feasibility of using each visualization technique in different characteristic flows developed in urban catchments, and to investigate the influence of rain intensity in velocity measurements as novel contribution. The LSPIV method, already validated in Naves et al. (2019a), is used as the reference technique in this analysis. Finally, the feasibility of these imaging techniques to measure runoff velocities in real field applications is discussed."*

**166 – Description of correlation matrix calculation is too brief. Need to help readers who may want to apply this technique themselves. Is this following what was done for other publications?**

**Response:**

The correlation matrix was computed using the Discrete Fourier transform (DFT) in the frequency domain, which is calculated using a fast Fourier transform (FFT). This is a common procedure to estimate particle displacement that is detailed in the bibliography of reference about PIV (Raffel et al., 2007; Adrian et al., 2011). The PIVLab software has this procedure implemented (Thieckle and Stamihus, 2014), where multi-pass window and deformation algorithm were used to improve the signal to noise ratio. The window size at the second pass achieves a higher spatial resolution. The searching area (SA) matches with the IA and 50% of overlapping was selected in all cases in the present work. The following text and references will be included in the text to complete the information about the correlation matrix calculation:

> *"Common procedures to estimate this particle displacement, and thus flow velocity, has been applied in the present work (Raffel et al., 2007; Adrian et al., 2011). The discrete Fourier transform (DFT), calculated using a fast Fourier transform (FFT), was used to compute the correlation matrix in the frequency domain. Moreover, two passes of a multi-pass window deformation algorithm were used in the present work, having the window size at the second pass to achieve a higher spatial resolution. The searching area (SA) matches with the IA and 50% of overlapping was selected in all cases in the present work. These procedures are included in most of the conventional PIV algorithms such as PIVLab (Thieckle and Stamihus 2014), or OpenPIV (Taylor et al. 2010)."*

**174 – not clear what you mean by 'which were investigated as an optimum'.**

**Response:**

These values were selected after some preliminary tests where the good performance detecting spurious vectors was checked. The sentence will be modified:

> *"After preliminary tests testing the performance detecting spurious vectors in the PIV results, the values of the two parameters of this filter were set to $\varepsilon = 0.15$ and threshold = 3. "*

**226 – 'This was to approach the conditions of worse devices . . .' is not clear.**

**Response:**

The sentence will be rewritten as follows:

> *"This simulates the FAR of worse installed devices that may serve as media source to measure urban runoff velocities in field applications, such as traffic or surveillance cameras following the ideas stated in Leitão et al. (2018)."*

279 – what type of flow specifically is in the area? The shallowness? Should be clarified.

**Response:**

Yes, it corresponds with the lowest depths analyzed. This will be clarified:

> *"The very low depths developed in this area also increases the variability of the mean velocities depending on the pair of frames analyzed."*

284 – should note that there is a degradation of quality with FAR, as expected.

**Response:**

We agree that it is a useful remark and it will be included.

> *"Finally, an expected degradation was noted when FAR is reduced, but within assumable ranges that make it possible to consider cameras with lower FAR as media source for field applications."*

317 - change phrasing. The current sentence uses a double negative. I think you mean that the unseeded technique is not able to measure the highest velocities, but i'm not sure.

**Response:**

Yes, we wanted to explain why unseeded techniques are not able to measure velocity when rain intensity and flow velocities are high. The text will be rewritten as follows:

> *"Then, the problems of unseeded techniques (SSIV and BIV) measuring velocities with high rain intensity are not produced because of a lack of tracers since, as can be observed in the videos provided in Naves et al. (2019b), the number of bubbles in that area increases with the rain intensity. These are caused by the erratic trajectory of the bubbles observed in the unseeded videos for high rain intensities due to the impact of raindrops on the water surface."*

322 – 'non-artificial' is again kind of a double negative. Just say what it is - the natural bubbles.

**Response:**

Thanks for the remark, 'non-artificial bubbles' will be substituted by 'natural bubbles'.

445 – do you mean that the problem is not trivial?

**Response:**

We referred to the difficulties of applying the seeded experiments methodology in real urban catchments during rain events, especially seeding particles. The sentence will be rewritten to avoid confusion:

> *"However, the ability of measuring extremely shallow flows where particles tend to be deposited, and their easy implementation without the need of adding artificial particles, make unseeded techniques worthy of future investigations as new source of runoff velocity data in urban catchments."*

24 – replace 'Specifically' with 'However'?

117 – 'estimate' is better than 'obtain' for this sentence.

117 – 'from the analysis of the images presented in the previous point' is not necessary.

224 – suggest 'typical of' rather than 'in consonance with'

261 – Acronym not introduced until next page (FAR)

325 – suggest 'prevents' rather than 'avoids'.

325 – 'from' obtaining

414 – should be Figure 10

**Response:**

Thanks again for the detailed review, we agree with the comments and these mistakes will be corrected in the revised version of the manuscript.

**References:**

Adrian, L., Adrian, R. J., and Westerweel, J.: Particle image velocimetry (No. 30), Cambridge University Press, 2011.

Leitão, J. P., Peña-Haro, S., Lüthi, B., Scheidegger, A., & de Vitry, M. M. (2018). Urban overland runoff velocity measurement with consumer-grade surveillance cameras and surface structure image velocimetry. Journal of Hydrology, 565, 791-804.

de Vitry, M. M., & Leitão, J. P. (2020). The potential of proxy water level measurements for calibrating urban pluvial flood models. Water Research, 115669.

Zhu, X., and Lipeme Kouyi, G.: An analysis of LSPIV-based surface velocity measurement techniques for stormwater detention basin management. Water Resour. Res., 55(2), 888-903, https://doi.org/10.1029/2018WR023813, 2019.

Pearce, S., Ljubičić, R., Peña-Haro, S., Perks, M., Tauro, F., Pizarro, A., ... & Paulus, G. (2020). An evaluation of image velocimetry techniques under low flow conditions and high seeding densities using Unmanned Aerial Systems. Remote Sensing, 12(2), 232. https://doi.org/10.3390/rs12020232

Raffel, M., Willert, C.E., Wereley, S.T., and Kompenhans, J.: Particle Image Velocimetry. Springer Berlin Heidelberg. 2007.

Raffel, M., Willert, C. E., Scarano, F., Kähler, C. J., Wereley, S. T., & Kompenhans, J. (2018). Particle image velocimetry: a practical guide. Springer.

Tauro, F., Petroselli, A., Porfiri, M., Giandomenico, L., Bernardi, G., Mele, F., ... & Grimaldi, S. (2016). A novel permanent gauge-cam station for surface-flow observations on the Tiber River. Geoscientific Instrumentation, Methods and Data Systems, 5(1), 241-251. https://doi.org/10.5194/gi-5-241-2016

Tauro, F., Petroselli, A., & Grimaldi, S. (2018). Optical sensing for stream flow observations: A review. Journal of Agricultural Engineering, 49(4), 199-206. https://doi.org/10.4081/jae.2018.836

Zhou, X., Doup, B., & Sun, X. (2013). Measurements of liquid-phase turbulence in gas–liquid two-phase flows using particle image velocimetry. Measurement Science and Technology, 24(12), 125303. https://doi.org/10.1088/0957-0233/24/12/125303

Thielicke, W., and Stamhuis, E.: PIVlab–towards user-friendly, affordable and accurate digital particle image velocimetry in Matlab, J. Open Res. Softw., 2(1), http://doi.org/10.5334/jors.bl, 2014.

Taylor, Z.J., Gurka, R., Kopp, G.A., and Liberzon, A. 2010. Long-duration timeresolved PIV to study unsteady aerodynamics. IEEE Transactions on Instrumentation and Measurement, 59(12): 3262–3269. doi:10.1109/TIM.2010.2047149.

---

## Author Comment (AC2) · 20 Jul 2020

**Response to reviewer 2**

**General comment**

Naves at all present a lab comparison of different particle imaging velocimetry techniques under (indoor) rainy conditions. I believe that analyzing this is a valuable addition to the scientific literature. I especially am happy that the authors have shown that not all PIV techniques perform equally under different rainy conditions, something very relevant when analyzing (urban) flood impact from video images. I commend the authors on their thorough effort of making the data on which they build their conclusions available to the public in true Open Science spirit.

I have a few minor issues with the paper in its current form, but am overall of the opinion that this paper should be published in HESS.

**Response:**

We would like to sincerely thank the reviewer for the time and effort invested in reviewing our manuscript and for the interest showed in our work. In the following, we provide detailed responses to reviewer's minor comments.

**Minor issues:**

The paper focusses on the application of urban flooding of streets and this is reflected in the literature cited. In river hydrology there are quite some papers also looking into using seeding for better LSPIV results. Multiple papers by Flavia Tauro and her team come to mind. Perhaps (but I'm not sure) adding these in the introduction would better frame the current research.

**Response:**

Thanks for the recommendation. We think that including papers related to river PIV applications will benefit the Introduction section. We will include the works of Tauro et al. (2016), Tauro et al. (2018), Pearce et al. (2020) and Manfreda et al. (2018) as follows:

> *"Imaging techniques are thus expanding in open and large-scale environments as non-intrusive methods for the characterization of surface velocity fields (Aberle et al., 2017), and their use is increasingly common in river monitoring (e.g. Tauro et al., 2016; Tauro et al., 2018; Manfreda et al., 2018; Pearce et al., 2020)."*

While the authors do make all their data available, and they do state which software packages they use for part of their analyses, it is impossible for me to check their results, since the code they use to generate their results is not shared. I would like to ask the authors to upload the code that generates the figures presented in the paper to Zenodo and cite it in the manuscript. This would also facilitate reproducing the result of this study, or expand on it.

**Response:**

As stated by the reviewer, our compromise with Open Science is clear, as can be seen in the experimental dataset cited in the manuscript where we made freely available our data

for others to be used in replicating our work or in conducting new research. In addition, we agree with the reviewer that sharing codes is a very recommendable practice to demonstrate more robustly and transparently the reliability of the results achieved, definitely benefiting research community. In this investigation, we have used existing and available codes during all the methodology, and the original code developed have been limited to facilitating computation of a considerable number of study cases and parameters by using simple loops.

The main functions and software used during the present work were: (1) 'fitgeotrans' and 'imwarp' Matlab functions to orthorectificate the frames analyzed, and 'rgb2gray' and simple comparisons using 'if' statements to apply sliding background and binarization during the preprocessing; (2) the command line script of PIVLab to compute the PIV cross-correlation (available at https://ch.mathworks.com/matlabcentral/fileexchange/27659-pivlab-particle-image-velocimetry-piv-tool?s_tid=mwa_osa_a); and (3) 'averf' and 'showf' functions from the 'pivmat' toolbox to visualize the mean velocity fields (available at http://www.fast.u-psud.fr/pivmat/).

In view of this, we consider that we have not developed any significant original code and we preferred to cite the sources within the text. However, we will be grateful to share our code on demand in the future if researchers need help to replicate our work or expand on it. As it does not appear in the manuscript, we will specify that the Matlab functions 'fitgeotrans' and 'imwarp' have been used to perform the orthorectification:

> *"To do this, frames were scaled and ortho-rectified using the known 2D coordinates of 28 and 24 reference surface points for each camera and the Matlab functions 'fitgeotrans' and 'imwarp'."*

The authors make use of the "jet" colormap for their figures, a choice that is known to results in figures that highlight differences not present in the data. (See among others https://www.jstor.org/stable/24862699?seq=1) Please switch to a different colormap. (this is a pet peeve of mine)

**Response:**

Thanks for sharing the reference, interesting issue. We have analyzed and compared 'jet' colormap results against 'haline' colormap from the mentioned reference. The results, which are included in the following figures, showed that in this particular case the 'jet' colormap do not include confusing data and, as stated in the reference mentioned by the reviewer, the sharp gradients of 'jet' colormap allow proximal colors to be distinguished, showing clearer the differences between techniques. In view of this, we have preferred to maintain 'jet' colormap to facilitate comparison with our previous work using LSPIV (cited in the text as Naves et al. 2019a). In any case, we thank the reviewer for the comment, and we will have this information into account for next communications.

[Figure]

Figure R1. Velocity fields representations for the rain intensity of 50 mm/h, the four techniques (rows) and the four study areas (columns), using "jet" colormap.

[Figure]

Figure R2. Velocity fields representations for the rain intensity of 50 mm/h, the four techniques (rows) and the four study areas (columns), using "haline" colormap.

Concluding: I really like the paper, the science, as presented, is sound although the actual claims cannot be verified without the software that generated their results shared alongside the paper.

**Response:**

Thanks again for your time and your interest in our work

**References:**

Pearce, S., Ljubičić, R., Peña-Haro, S., Perks, M., Tauro, F., Pizarro, A., ... & Paulus, G. (2020). An evaluation of image velocimetry techniques under low flow conditions and high seeding densities using Unmanned Aerial Systems. Remote Sensing, 12(2), 232. https://doi.org/10.3390/rs12020232

Tauro, F., Petroselli, A., Porfiri, M., Giandomenico, L., Bernardi, G., Mele, F., ... & Grimaldi, S. (2016). A novel permanent gauge-cam station for surface-flow observations on the Tiber River. Geoscientific Instrumentation, Methods and Data Systems, 5(1), 241-251. https://doi.org/10.5194/gi-5-241-2016

Tauro, F., Petroselli, A., & Grimaldi, S. (2018). Optical sensing for stream flow observations: A review. Journal of Agricultural Engineering, 49(4), 199-206. https://doi.org/10.4081/jae.2018.836

Manfreda, S., McCabe, M. F., Miller, P. E., Lucas, R., Pajuelo Madrigal, V., Mallinis, G., ... & Müllerová, J. (2018). On the use of unmanned aerial systems for environmental monitoring. Remote sensing, 10(4), 641.

---

## Author Response (AR1)

**COMMENTS TO EDITOR**

Your manuscript has been reviewed by two referees. Both referees provided very positive feedback highlighting the value of your contribution, and both suggested that some minor revisions be made. One more "major" criticism, though, was that your manuscript may be "too positive about the results", and that your optimism may be "related to other studies or results that are not included in the current paper" (I am quoting one referee, here). I do agree with the referee that the main conclusions of your manuscript regarding unseeded methods should be based on data/information reported in your manuscript. That being said, I can see from your responses to the referees that you have already identified where – in the Results and Discussion sections – you will need to make text changes to nuance some of your statements/conclusions.

**Response:**

This document contains the replies to the comments of the article hess-2020-136 titled "Assessing different imaging velocimetry techniques to measure shallow runoff velocities during rain events using an urban drainage physical model" by Juan Naves et al.

First, we want to thank the editor and the reviewers for their positive evaluation of our work and the very constructive suggestions to improve our manuscript. We sincerely believe that the comments and suggestions have significantly contributed to improve the quality of the manuscript. In this document, we provide detailed responses to each reviewer's comment indicating all relevant changes made in the manuscript, and a marked-up version of the manuscript to facilitate a concise review process.

As reviewers and editor may note in the revised version of the manuscript, we have changed the nomenclature of the 'SSIV technique' by 'LSPIVu technique'. This change is motivated by a request received by mail from Salvador Peña when the HESS discussion was closed. S. Peña works at the company that developed the SSIV method (Photrack AG - http://photrack.ch) appearing as author in Leitão et al. (2018), Lüthi et al. (2014) and Hansen et al. (2017), which were referenced in the present study. He stated that the SSIV methodology was not clearly explained in their collaboration in Leitão et al. (2018) because it is a proprietary software, but it slightly differs from our procedure and, although we have not transgressed any intellectual property, he kindly asked us to use other nomenclature for avoiding confusion to their potential customers, since photrack is economically exploiting the SSIV methodology. Therefore, as our intention of using 'SSIV' was to refer to the name given by the first application of LSPIV for obtaining surface velocities form urban catchments without using artificial particles and we do not want to cause prejudice, we have decided to change this nomenclature from SSIV to LSPIVu within the manuscript. In the following, we have included the mail received from S. Peña and the mail that we sent to the editor with which we confirmed that changing the nomenclature at this stage of the revision was within the procedures of the journal:

Dear Juan Naves,
My name is Salvador Peña-Haro and I work at photrack where we have developed the SSIV
methodology.

Today I came across your publication at HESS "Assessing different imaging velocimetry techniques to
measure shallow runoff velocities during rain events using an urban drainage physical model" and I
was surprised to see the SSIV method mentioned.

The description of the SSIV method in your article is very different from our methodology, probably
because you have based your application on the paper of Leitao et al., 2018 where the method is
not fully explained since it is proprietary. Although in your paper you developed a correct
methodology using LSPIV to determine surface runoff velocity and you have not transgressed any
intellectual property, I like to kindly ask you to change the name of your method to avoid confusion
with the SSIV method given the differences between both. I believe these changes are very easy to
implement since they are only related to the name SSIV. Of course you can refer -and we will be
happy if you do- to Leitao et al., 2018.

As the open discussion at HESS is closed, I'm not able to comment this issue on the journal website.
Kind regards,

Salvador Peña-Haro
CTO
photrack AG

Dear Editor Genevieve Ali,

We are finishing the review of the HESS manuscript hess-2020-136 'Assessing different imaging
velocimetry techniques to measure shallow runoff velocities during rain events using an urban
drainage physical model' by Naves et al., and we will upload the requested files with the changes
demanded as soon as possible.

Regarding the above manuscript, we would like to make an enquiry about a request that we have
recently received from Salvador Peña (**Photrack AG, Flow measurements**), asking us to change the
nomenclature used for one of the imaging techniques included within our article. Below is included
the e-mail received from S. Peña.

Following S. Peña comments, in our HESS manuscript we are using 'SSIV' (Surface Structure Image
Velocimetry) to name the application of LSPIV technique without artificial particles following the

paper of Leitao et al. (2018) (https://doi.org/10.1016/j.jhydrol.2018.09.001). In response to this mail, we arranged a meeting with Salvador where he explained that they named SSIV to a specific imaging processing methodology that was developed by his company, is a patented method and is being economical exploited by photrack. SSIV methodology was not clearly explained in their collaboration in Leitão et al. (2018) because it is proprietary, but according to them it slightly differs from our procedure and, although we have not transgressed any intellectual property, they kindly ask us to use other nomenclature for avoiding confusion to their potential customers.

The intention of using 'SSIV' was to refer to the name given by the first application of LSPIV for obtaining surface velocities form urban catchments without using artificial particles, and we do not wanted in any way to cause prejudice to anyone. Therefore, considering that the change of nomenclature within our manuscript is easy to implement and it does not affect to any scientific contribution, we are willing to change the name of the technique in the revised version of the manuscript and explaining the motivation.

However, as the request has been formulated once the HESS discussion was closed, we would like to ensure that doing this is within the procedures of the journal and if additional data (like the mail received) will be specifically required in the Author's Response document

 Thanks for your time and consideration.

 Best regards,

 Juan

In addition, we have slightly modified the text to adapt the content to the new nomenclature and note that our LSPIVu technique is also a LSPIV-based method (like SSIV) to obtain runoff velocities without using artificial particles. Pg 2 Ln 55 and Pg 6 Ln 33:

*"In addition, LSPIV-based methods can be applied to determine runoff velocities without the presence of particles such as in Leitão et al. (2018), where a method called Surface Structure Image Velocimetry (Lüthi et al., 2014; Hansen et al., 2017) introduces some improvements based on image pre-processing analysis to measure shallow flows in a flood experimental facility."*

*"Then, a LSPIV-based method named as LSPIVu in this work and the BIV technique were used to obtain velocity fields from the unseeded and LED-illuminated experiments. LSPIVu is inspired in the non-open SSIV procedure employed in Leitao et al. (2018) and uses a SLB image pre-processing to remove the background from the analysis and satisfactorily trace the movement of air bubbles and surface water reflections generated by raindrops."*

**COMMENTS TO REVIEWER 1**

**General comments**

This study assesses four image based velocimetry techniques for measuring water velocity in shallow flows as would be observed in overland flows over paved surfaces during rainfall events. The problem is physically challenging, and the authors use a dedicated lab setup to assess these techniques with an eye to evaluating their potential for more difficult and varied conditions in the field. The work is derived from a larger project that has led to a number of significant publications over the last couple of years and is a direct extension of a 2019 paper that validated one of the approaches used (LSPIV). The validated approach is used as the reference condition for the current paper. The paper overall was well written and the methods appeared to be suitable for assessing the other velocimetry techniques.

**Response:**

We appreciate the reviewer for the time invested in reviewing our manuscript and for the positive evaluation of our work. We are grateful for the detailed review and believe that the constructive comments and suggestions will lead to a deeper and clearer analysis of the results presented in this article, contributing to significantly improve the quality of the manuscript. In this document, we present the responses to the reviewer's comments indicating how and where these were addressed in the revised version of the manuscript.

Despite the quality of the work, the authors in my opinion are too positive about the results. In looking at the results from my reviewer's perspective, it appears that the unseeded techniques are not suitable for measuring velocities in shallow flows. Even in relatively straight flows with low precipitation, there is an offset between the unseeded techniques and the LSPIV results that is not well explained. It is not clear to me how the magnitude of this offset could be predicted without controlled tests. As the precipitation intensity increases, the error in the unseeded techniques increases to the point where the results are no longer even correlated with the validated technique. In these conditions I would argue that the unseeded techniques are simply not suitable. Despite this, many of the statements in the discussion and conclusions are quite positive about the techniques. The optimism seems to be related to other studies or results that are not included in the current paper. Something needs to be adjusted, either by including those results (maybe cases without any precipitation at all?) or by drawing sharper lines about which techniques are reliable in different conditions.

**Response:**

Based on the comment of the reviewer, we have reread the manuscript and we agree that the discussion and conclusions appear too positive considering results presented for unseeded methods BIV and LSPIVu (as explained in the comment to the editor, we have changed the nomenclature of SSIV by LSPIVu). This may lead journal readers to confusion if a clear and contextualized interpretation of the results is not included. The positiveness showed is due to the great potential of unseeded techniques as a tool to obtain runoff velocity data from media sources commonly available in urban environments, such as surveillance cameras, traffic cameras, or

even social media. From our point of view, this is a novel and powerful data source with a great potential to solve the current lack of surface runoff velocity data, which is key in the proper calibration of the increasingly more accurate 2D-1D dual urban drainage models that are currently being developed. The use of these data sources has been recently introduced in the field of urban drainage (Leitão et al. 2018, de Vitry et al. 2020). In addition, in contrast with the increasingly more common application of visualization techniques to rivers monitorization (Pearce et al. 2020, Tauro et al. 2016, Tauro et al. 2018), their use for urban runoff measurements is still limited to some initial applications on the analysis of the velocities in a stormwater storage facility (Zhu et al., 2019) and in simulated urban floods (Leitão et al. 2018), both without precipitation. Therefore, this is the first work where the influence of raindrop interference with the recorded images and their impact in the measurement of velocities is analyzed, and therefore is still much room for improvement in this novel implementation.

In this context, we consider as positive the results obtained from unseeded techniques for low rain intensities where the results correlate with LSPIV results, since two different techniques are being used. The different density and size of bubbles and artificial fluorescent particles explains the offset found for low rain intensities, since tracers are affected in different degrees by raindrop impacts and may be transported at different velocities. The gap obtained in this study between seeded and unseeded experiments may be thus interpreted as an indicator of the uncertainties that may appear when using visualization techniques in rainy conditions depending on the type of tracer.  As commented by the reviewer, we think that the prediction of the magnitude of this offset is currently challenging, so further investigation on how rain impacts the transport of possible tracers appears as an interesting research line to reduce uncertainties in runoff velocity measurements. Finally, the different tracers analyzed also explain the different behavior of seeded and unseeded techniques when the rain intensity is increased. While raindrop impacts produce fast and random changes of position in bubbles, that are incremented with rain intensity, the higher density of fluorescent particles confers themselves inertia to avoid such sudden movements. We think that this great affection of raindrops in unseeded techniques, leading to erroneous results for high rain intensities, is an important result of the present work and we agree that this should have been clearer presented in the text.

We have thus revised the results and discussion and conclusions chapters to clarify these points by, as suggested by the reviewer, drawing sharper lines between the performance of each methods, explaining deeply the offset obtained between seeded and unseeded techniques, and discussing the potential and possible challenges of visualization techniques to measure urban runoff velocities. Among other little modifications, the following sentences have been added in the Results and discussion and Conclusions sections to clearly specify that unseeded methods are not working for high intensities (Pg 13, Ln 335; Pg 19, Ln 423; and Pg 22, Ln 497):

*"However, the velocity fields obtained for rain intensities of 50 and 80 mm h$^{-1}$ showed that both the LSPIVu as well as the BIV techniques resulted in erroneous velocity distributions, being more affected the areas where greater velocities are developed"*

*"The use of bubbles as tracers gives unseeded techniques an opportunity to measure velocities in extremely shallow flows where particles tend to be deposited. However, LSPIVu and BIV are more affected by the impact of raindrops leading to erroneous results for high rain intensities, especially for high velocity flows"*

*"Unseeded techniques are highly affected by raindrop impacts. First, the gap between seeded and unseeded techniques is reduced as the rain intensity is increased, so rain intensity should be also considered to determine the velocity index for estimating depth-average velocities. Then, raindrop impacts also produce fast and random changes of position of the bubbles used as tracers, leading to erroneous velocities for high rain intensities. However, the ability of measuring extremely shallow flows where particles tend to be deposited, and their easy implementation without the need of adding artificial particles, make unseeded techniques worthy of future investigations as new source of runoff velocity data in urban catchments."*

The explanation of the offset between seeded and unseeded experiments has been completed in Section 3.2 (Pg 13, Ln 326) and stated in conclusions (Pg 22, Ln 487) as follows:

*"All visualization techniques presented a similar velocity distribution for the lowest rain intensity (first row), although an offset of approximately 0.05 m s-1 was obtained for the unseeded techniques. This offset is produced because the different tracers used in seeded and unseeded experiments, which are affected in different degrees by raindrop impacts and may be transported at different velocities. Considering the novel application of these techniques in presence of rain, it can be deduced that all techniques obtained a good performance for 30 mm h$^{-1}$ rainfall and that lower velocity indexes are required in the case of unseeded techniques to convert the results to depth-averaged velocities, as observed in previous references (Leitão et al., 2018; Martins et al., 2018; Naves et al., 2019a)"*

*"Both seeded and unseeded techniques provide suitable velocity distributions for lower rain intensities in case of unidirectional flows, observing an offset of approximately 0.05 m s-1 between them. This offset is a consequence of the different tracers used in seeded and unseeded experiments, which are affected in different degrees by raindrop impacts and may be transported at different velocities. Lower velocity indexes are thus required in the case of unseeded techniques to convert the results to depth-averaged velocities. In case of more complex flows, unseeded techniques are not able to adequately measure since bubbles have difficulties to follow the runoff generated."*

**Questions and comments:**

148 – Is the LSPIVb procedure significantly different than the LSPIV? In reading the methods I thought that the results might arrive at the same point as each requires a threshold, one applied to the difference, the other to the base images and then the difference is then

calculated. The results also show that they are nearly the same. The point of the LSPIVb analysis is not emphasized in the paper. What is the motivation for evaluating this technique? It is not really discussed in the results or appear in the conclusions to a significant extent. Does it 'better remove background and shadows. . .''? Should other people use it instead of the regular LSPIV?

**Response:**

This is an interesting comment of the reviewer. It is true that preprocessing of both LSPIV and LSPIVb depends on only one threshold, but the different purpose of these thresholds leads to different preprocessed images. First, the sliding background preprocess was applied in the LSPIV technique to remove all elements that remain fixed between two consecutive frames, including the road surface, other elements of the physical model, and immobile particles. The threshold used in this case corresponds a percentage of the grey value to consider that an element does not move since, although the pixel correspond to a immobile element, this value may slightly vary because of variations on water surface or raindrops interferences. In contrast, the binarization performed for LSPIVb technique seeks mainly to isolate the brightest pixels, which in this case will correspond with the fluorescence particles used as tracers. Then, we also remove deposited and immobile particles with the sliding background filter, but the binarization makes the previous margin of gray value used in LSPIV unnecessary.

Binarization is used in PIV studies (e.g. Zhou et al. 2013) to remove the remaining noise in raw PIV images, resulting in images where all the particles have the same intensity and thus have equal contribution to the correlation function. However, this preprocessing technique would lead to increase measurement uncertainty if the threshold value is not properly addressed (Raffel et al. 2018). The motivation of including binarization in seeded experiments (LSPIVb technique) was firstly consistency with the unseeded techniques found in the literature. While LSPIVu seeks to remove immobile features from frames through sliding background and analyze the movement of all other elements, BIV seeks to analyze only the movement of bubbles, which are identified as the brightest elements in the images, removing the rest of features from the images. When we applied these preprocessing procedures to seeded experiments images, we observed that the processed images were slightly different and the binarization (in LSPIVb technique) reduced the number of particles to analyze in the images. For example, this can be observed in the following figure (Figure R1), where the same frame obtained from seeded videos was preprocessed following the procedures for LSPIV and LSPIVb respectively.

[Figure]

Figure R1. Preprocessed frame for LSPIV and LSPIVb imaging techniques.

In view of the differences between the images and the use of binarization in the literature, we decided to include LSPIVb technique to investigate the influence of binarization in the analysis of seeded experiments. As noted by the reviewer, the achieved results with both seeded techniques are very similar with slightly higher velocities obtained by techniques that use binarization. This similarity indicates that, except particles and bubbles, cameras did not record many other moving elements that disturb the results, so binarization does not include significant benefits in these experiments. Another interesting difference observed is that techniques including binarization resulted in noisier velocity results. This may be due to the fact that, if the binarization is applied, the sliding background filter may remove parts of tracers in motion that are overlapped in consecutive frames since no different grey values are considered, which also might explain the slightly higher velocities obtained. This indicates that binarization, which may be useful to isolate tracers if seeded experiments are performed with natural or regular artificial illumination, should be used with care in future applications and it is not recommended if the non-binarized images results in good correlations.

A detailed explanation of the motivation of evaluating LSPIVb and a more extended discussion about the similar results obtained using LSPIV and LSPIVb have been added to Methods and Results sections for a better comprehension of the achieved results. Specifically, the following text has been added to the methodology (Sections 2.2 and 2.2.1 respectively) to clarify the motivation of LSPIVb (Pg 6, Ln 138 and Pg 7, Ln 163):

*"Finally, a slight variation of the LSPIV methodology named LSPIVb was implemented to investigate the influence of binarization pre-process also in the analysis of seeded UV experiments. This strategy seeks to isolate the brightest pixels, which in this case correspond with the fluorescence particles, to ensure that other elements such as bubbles or water reflections are not interfering in the PIV analysis."*

*"This filter ensures that only the fluorescent particles are being considering in the PIV cross-correlation, preventing possible small interferences that bubbles or water reflections may produce despite the special illumination* (Zhou et al. 2013)*."*

In addition, the similar results obtained by LSPIV and LSPIVb have been discussed in Results and discussion section as follows (Pg 18, Ln 407):

*"Then, the velocity fields showed very similar results between LSPIV and LSPIVb and between LSPIVu and BIV with slightly higher velocities measured by methods using binarization pre-processing (LSPIVb and BIV). This similarity indicates that, except particles and bubbles, cameras did not record many other moving elements that disturb the results, so binarization does not includes significant benefits in these experiments. In addition, it has been observed that techniques that include binarization result in noisier velocity results (see velocity fields for 30 and 80 mm h-1 in the supplementary information). This may be due to the fact that, if binarization is applied, the sliding background filter may remove parts of tracers in motion that are overlapped in consecutive frames since no different grey values are considered, which also might explain the slightly higher velocities obtained. Therefore, this filter should be used with care in future applications if it would be necessary to isolate tracers from other mobile elements."*

Finally, an additional point has been added to conclusions (Pg 23, Ln 504):

*"The similarity found between LSPIV and LSPIVb and between LSPIVu and BIV indicates that binarization preprocessing has not significant benefits in these experiments since cameras did not record moving elements that significantly disturb the results. In addition, it has been found that this procedure lead to noisier results, so binarization should be used with care in future applications if it would be necessary to isolate tracers from other mobile elements."*

**Minor issues:**

16 - complex sentence. should split into one about the natural tracers and the second about the raindrop impacts.

**Response:**

The sentence has been split and slightly modified to simplify the text (Pg 1, Ln 17):

*"First, the use of naturally-generated bubbles and water shadows and glares as tracers allows the unseeded techniques (LSPIVu and BIV) to measure extremely shallow flows. However, these techniques are more affected by raindrop impacts, which even lead to erroneous velocities in the case of high rain intensities."*

47 – sentence starting with 'For instance' is not clear to me. Should be rewritten in a more direct style.

**Response:**

The sentence have been rewritten in a more direct style as follows (Pg 2, Ln 52):

*"Zhu et al. (2019) achieved errors below 14% using this technique in a full-scale stormwater detention basin, although in some bordering points these could rise up to 44%."*

61 – sentence starting with 'That study' is too complex. Should be split into two ideas.

**Response:**

The sentence has been rewritten for a better understanding (Pg 3, Ln 67):

*"The presence of raindrops in the experiments can generate disturbances in the water surface and also interfere in the visualization of images, so that study used UV illumination and fluorescent particles as artificial tracers to satisfactorily address these issues."*

87 and 106 – is Naves et al 2019b an archive? Data availability should be clarified.

**Response:**

Yes, the reference is from a dataset published by the authors in the open access repository Zenodo. This includes videos, images, and related information to replicate our study or produce new results. This has been specified within the text as follows (Pg 4, Ln 96 and Pg 4, Ln 117):

*"The freely available experimental dataset (Naves et al. 2019b) described in Naves et al. (2020b) was used in this study for the assessment of different imaging velocimetry techniques.*

*"Examples of these images obtained from UV seeded and LED unseeded experiments, which are openly available for others to use in the dataset published by the authors (Naves et al., 2019b), are included in Fig. 2."*

101 - Best to say what was done step by step. e.g. Videos were recorded at 4K resolution and 25Hz. 1500 frames (eq. to 60 s) were extracted from the longer recording for analysis.

**Response:**

As recommended, the sentences have been rewritten to clarify the methodology (Pg 4, Ln 111):

*"During the experiments, videos were recorded at 4K resolution and 25 Hz. 1500 frames of steady flow (equivalent to 60 s) were then extracted from the longer recording and processed for analysis. To do this, frames were scaled and ortho-rectified using the known 2D coordinates of 28 and 24 reference surface points for each camera and the Matlab functions 'fitgeotrans' and 'imwarp'. Finally, the reference points placed in the intersection between the recorded areas of each camera were used to crop and join the images, resulting in raw images where 1 pixel corresponds to 1 mm in real-world coordinates."*

120 – so all particles are assumed to be moving? Is this realistic? Is there a velocity threshold?

**Response:**

Some of the particles used as tracers may settle on the road surface due to the extreme low depths developed in some areas of the road surface and to the rugosity of the concrete surface. While the rest of particles follow the runoff generated, these particles appear immobile in the images recorded. This can lead to erroneous velocity results when the PIV cross-correlation is performed, because the null velocity of these particles can reduce the mean velocity of the particles of a determined interrogation area. The sliding background preprocess avoids this issue removing the immobile particles and focusing the analysis on the particles that are being transported by the flow. In addition, this procedure does not produce any velocity threshold in the results since only the particles that remain immobile between two consecutive frames are removed from the analysis, as can be also checked in the previous work published by the authors (Naves et al. 2019a). However, as stated within the text, the seeded techniques are not able to measure velocities in areas with extreme shallow flows (Area 2) because the artificial particles cannot be transported, and unseeded techniques appear as suitable tools to be further explorer for these conditions.

The sentence has been rewritten to clearly explain the removal of immobile particles (Pg 5, Ln 130):

> *"That methodology requires pre-processing of images through a sliding background (SLB), which eliminates the background of the images and particles that remain immobile between frames. These particles, which are deposited due to the extreme low depths developed and the rugosity of the concrete surface, should be removed to avoid that the null velocities resulted from them condition the PIV analysis."*

156 – restatement of the aim/objective. Not necessary in the methods.

180 – again repetition of aim, but shouldn't be necessary.

183 – novelty should be addressed in intro with aim and objectives.

193 – more repetition of the aim

195 – I think that the reference technique statement should also be used as a scoping statement at the end of the introduction with the aim/objectives. Mixing it in here reduces the clarity of what is being done and what the starting point for the new contribution is.

**Response:**

We agree with the reviewer. The final statement of the introduction section has been completed to include clearer the aim, the novelties, and the starting point of the new contribution (Pg 3, Ln 79). This content has been removed from the rest of the manuscript where, as noted by the reviewer, it is not necessary.

*"Therefore, experimental videos of the overland flow generated by three different rain intensities, under laboratory-controlled conditions and recorded with and without artificial particles, are used in this study to comparatively assess the performance of different seeded and unseeded imaging velocimetry techniques under rainy conditions. First, the sensitivity of the velocity results to the analysis parameters is investigated in order to test the robustness of each method. Then, the resulting velocity fields are compared to analyze the feasibility of using each visualization technique in different characteristic flows developed in urban catchments, and to investigate the influence of rain intensity in velocity measurements as novel contribution. The LSPIV procedure, already validated in Naves et al. (2019a), is used as the reference technique in this analysis. Finally, the feasibility of these imaging techniques to measure runoff velocities in real field applications is discussed."*

166 – Description of correlation matrix calculation is too brief. Need to help readers who may want to apply this technique themselves. Is this following what was done for other publications?

**Response:**

The correlation matrix was computed using the Discrete Fourier transform (DFT) in the frequency domain, which is calculated using a fast Fourier transform (FFT). This is a common procedure to estimate particle displacement that is detailed in the bibliography of reference about PIV (Raffel et al., 2007; Adrian et al., 2011). The PIVLab software has this procedure implemented (Thieckle and Stamihus, 2014), where multi-pass window and deformation algorithm were used to improve the signal to noise ratio. The window size at the second pass achieves a higher spatial resolution. The searching area (SA) matches with the IA and 50% of overlapping was selected in all cases in the present work. The following text and references have been included in the text to complete the information about the correlation matrix calculation (Pg 7, Ln 184):

*"Common procedures to estimate this particle displacement, and thus flow velocity, has been applied in the present work (Raffel et al., 2007; Adrian et al., 2011). The discrete Fourier transform (DFT), calculated using a fast Fourier transform (FFT), was used to compute the correlation matrix in the frequency domain. Moreover, two passes of a multi-pass window deformation algorithm were used in the present work, having the window size at the second pass to achieve a higher spatial resolution. The searching area (SA) matches with the IA and 50% of overlapping was selected in all cases in the present work. These procedures are included in most of the conventional PIV algorithms such as PIVLab (Thieckle and Stamihus 2014), or OpenPIV (Taylor et al. 2010)."*

174 – not clear what you mean by 'which were investigated as an optimum'.

**Response:**

These values were selected after some preliminary tests where the good performance detecting spurious vectors was checked. The sentence has been modified (Pg 8, Ln 196):

*"After preliminary tests assessing the performance detecting spurious vectors in the PIV results, the values of the two parameters of this filter were set to ε = 0.15 and threshold = 3. "*

226 – 'This was to approach the conditions of worse devices . . .' is not clear.

**Response:**

The sentence has been rewritten as follows (Pg 10, Ln 251):

*"This simulates the FAR of some already installed devices that may serve as media source to measure urban runoff velocities in field applications, such as traffic or surveillance cameras following the ideas stated in Leitão et al. (2018)."*

279 – what type of flow specifically is in the area? The shallowness? Should be clarified.

**Response:**

Yes, it corresponds with the lowest depths analyzed. This has been clarified (Pg 12, Ln 305):

*"The very low depths developed in this area also increases the variability of the mean velocities depending on the pair of frames analyzed."*

284 – should note that there is a degradation of quality with FAR, as expected.

**Response:**

We agree that it is a useful remark and it has been included as follow (Pg 13, Ln 311):

*"Finally, an expected degradation was noted when FAR is reduced, but within assumable ranges that make it possible to consider cameras with lower FAR as media source for field applications."*

317 - change phrasing. The current sentence uses a double negative. I think you mean that the unseeded technique is not able to measure the highest velocities, but i'm not sure.

**Response:**

Yes, we wanted to explain why unseeded techniques are not able to measure velocity when rain intensity and flow velocities are high. The text has been rewritten as follows (Pg 14, Ln 350):

*"Then, the problems of unseeded techniques (LSPIVu and BIV) measuring velocities with high rain intensity are not produced because of a lack of tracers since, as can be observed in the videos provided in Naves et al. (2019b), the number of bubbles in that area increases with the rain intensity. These are caused by the erratic trajectory of the bubbles observed in the unseeded videos for high rain intensities due to the impact of raindrops on the water surface."*

322 – 'non-artificial' is again kind of a double negative. Just say what it is - the natural bubbles.

**Response:**

Thanks for the remark, 'non-artificial bubbles' has been substituted by 'natural bubbles' (Pg 15, Ln 356).

445 – do you mean that the problem is not trivial?

**Response:**

We referred to the difficulties of applying the seeded experiments methodology in real urban catchments during rain events, especially seeding particles. The sentence has been rewritten to avoid confusion as follows (Pg 22, Ln 500):

> *"However, the ability of measuring extremely shallow flows where particles tend to be deposited, and their easy implementation without the need of adding artificial particles, make unseeded techniques worthy of future investigations as new source of runoff velocity data in urban catchments."*

24 – replace 'Specifically' with 'However'?

117 – 'estimate' is better than 'obtain' for this sentence.

117 – 'from the analysis of the images presented in the previous point' is not necessary.

224 – suggest 'typical of' rather than 'in consonance with'

261 – Acronym not introduced until next page (FAR)

325 – suggest 'prevents' rather than 'avoids'.

325 – 'from' obtaining

414 – should be Figure 10

**Response:**

Thanks again for the detailed review, we agree with the comments and these mistakes have been corrected in the revised version of the manuscript.

**COMMENTS TO REVIEWER 2**

**General comment**

Naves at all present a lab comparison of different particle imaging velocimetry techniques under (indoor) rainy conditions. I believe that analyzing this is a valuable addition to the scientific literature. I especially am happy that the authors have shown that not all PIV techniques perform equally under different rainy conditions, something very relevant when analyzing (urban) flood impact from video images. I commend the authors on their thorough effort of making the data on which they build their conclusions available to the public in true Open Science spirit.

I have a few minor issues with the paper in its current form, but am overall of the opinion that this paper should be published in HESS.

**Response:**

We would like to sincerely thank the reviewer for the time and effort invested in reviewing our manuscript and for the interest showed in our work. In the following, we provide detailed responses to reviewer's minor comments.

**Minor issues:**

The paper focusses on the application of urban flooding of streets and this is reflected in the literature cited. In river hydrology there are quite some papers also looking into using seeding for better LSPIV results. Multiple papers by Flavia Tauro and her team come to mind. Perhaps (but I'm not sure) adding these in the introduction would better frame the current research.

**Response:**

Thanks for the recommendation. We think that including papers related to river PIV applications will benefit the Introduction section. We have included the works of Tauro et al. (2016), Tauro et al. (2018), Pearce et al. (2020) and Manfreda et al. (2018) as follows (Pg 2, Ln 44):

*"Imaging techniques are thus expanding in open and large-scale environments as non-intrusive methods for the characterization of surface velocity fields (Aberle et al., 2017), and their use is increasingly common in river monitoring (e.g. Tauro et al., 2016; Tauro et al., 2018; Manfreda et al., 2018; Pearce et al., 2020)."*

While the authors do make all their data available, and they do state which software packages they use for part of their analyses, it is impossible for me to check their results, since the code they use to generate their results is not shared. I would like to ask the authors to upload the code that generates the figures presented in the paper to Zenodo and cite it in the manuscript. This would also facilitate reproducing the result of this study, or expand on it.

**Response:**

As stated by the reviewer, our compromise with Open Science is clear, as can be seen in the experimental dataset cited in the manuscript where we made freely available our data for others to be used in replicating our work or in conducting new research. In addition, we agree with the reviewer that sharing codes is a very recommendable practice to demonstrate more robustly and transparently the reliability of the results achieved, definitely benefiting research community. In this investigation, we have used existing and available codes during all the methodology, and the original code developed have been limited to facilitating computation of a considerable number of study cases and parameters by using simple loops.

The main functions and software used during the present work were: (1) 'fitgeotrans' and 'imwarp' Matlab functions to orthorectificate the frames analyzed, and 'rgb2gray' and simple comparisons using 'if' statements to apply sliding background and binarization during the

preprocessing; (2) the command line script of PIVLab to compute the PIV cross-correlation (available at https://ch.mathworks.com/matlabcentral/fileexchange/27659-pivlab-particle-image-velocimetry-piv-tool?s_tid=mwa_osa_a); and (3) 'averf' and 'showf' functions from the 'pivmat' toolbox to visualize the mean velocity fields (available at http://www.fast.u-psud.fr/pivmat/).

In view of this, we consider that we have not developed any significant original code and we preferred to cite the sources within the text. However, we will be grateful to share our code on demand in the future if researchers need help to replicate our work or expand on it. As it did not appear in the manuscript, we have specified that the Matlab functions 'fitgeotrans' and 'imwarp' have been used to perform the orthorectification (Pg 4, Ln 113):

*"To do this, frames were scaled and ortho-rectified using the known 2D coordinates of 28 and 24 reference surface points for each camera and the Matlab functions 'fitgeotrans' and 'imwarp'."*

The authors make use of the "jet" colormap for their figures, a choice that is known to results in figures that highlight differences not present in the data. (See among others https://www.jstor.org/stable/24862699?seq=1) Please switch to a different colormap. (this is a pet peeve of mine)

**Response:**

Thanks for sharing the reference, interesting issue. We have analyzed and compared 'jet' colormap results against 'haline' colormap from the mentioned reference. The results, which are included in the following figures, showed that in this particular case the 'jet' colormap do not include confusing data and, as stated in the reference mentioned by the reviewer, the sharp gradients of 'jet' colormap allow proximal colors to be distinguished, showing clearer the differences between techniques. In view of this, we have preferred to maintain 'jet' colormap to facilitate comparison with our previous work using LSPIV (cited in the text as Naves et al. 2019a). In any case, we thank the reviewer for the comment, and we will have this information into account for next communications.

[Figure]

Figure R1. Velocity fields representations for the rain intensity of 50 mm/h, the four techniques (rows) and the four study areas (columns), using "jet" colormap.

[Figure]

Figure R2. Velocity fields representations for the rain intensity of 50 mm/h, the four techniques (rows) and the four study areas (columns), using "haline" colormap.

Concluding: I really like the paper, the science, as presented, is sound although the actual claims cannot be verified without the software that generated their results shared alongside the paper.

**Response:**

Thanks again for your time and your interest in our work

**References:**

[revised manuscript text omitted]

---

## Author Response (AR2)

**COMMENTS TO EDITOR**

I have now heard from two reviewers regarding your revised manuscript. One reviewer (who had seen the original version of your manuscript) is satisfied with the modifications you have made. Another reviewer (who had not reviewed your original manuscript) also finds merit in your work but would like you to address a few points before your manuscript can be ready for publication, particularly one regarding the limits/challenges associated with your methodology. I agree that a more "balanced" discussion of the advantages and disadvantages of your methodology will be helpful to the readers of HESS. I am therefore returning your manuscript for minor revisions.

**Response:**

   This document contains the replies to the comments of the article hess-2020-136 titled "Assessing different imaging velocimetry techniques to measure shallow runoff velocities during rain events using an urban drainage physical model" by Juan Naves et al. In this document, we respond the comments that have arisen in this new review and indicate the relevant changes made in the manuscript. These changes focus mainly on clarifying the advantages and disadvantages of each method. Finally, we would like to thank the editor and the reviewers for their accurate and fair review and for all the time invested in improving our manuscript.

**COMMENTS TO REVIEWER 1**

The authors have responded to all my comments and I have none further. I thank the authors for considering my comments and responding in a detailed way, with modifications to the paper where appropriate. I think that the facility is an important one for testing these types of techniques and that the paper should be a valuable contribution to the field.

**Response:**

   Thank you again for your time and helpful comments.

**COMMENTS TO REVIEWER 3**

The review relates to the revised version of the manuscript, and it considers the answers that had been given to the previous review. The well-prepared manuscript compares the performance of different imaging velocimetry techniques for overland flow velocity estimation under the influence of raindrops by means of controlled pilot-scale experiments. It represents a valuable scientific contribution in the field, in particular as the study comes with a comprehensive data set, i.e. findings that are based on solid experimental research. Well-done and very appreciated!

**Response:**

   We would like to thank the reviewer for the interest showed on our work and the positive evaluation. We think that the external point of view of the reviewer has led to fair, concise, and useful comments. We have done our best to address them and improve the discussion of our results, so that it is helpful and clear for future laboratory and real-world applications.

**R3C1:** Revised manuscript follows a classical structure, provides (considering references to other publications of the group) sufficient insights in the design of the experiments, presents illustrative results, followed by a discussion slightly biased towards the benefits, avoiding the challenges. The "all-too-positive" tenor, that had been criticized by a previous reviewer (which I agree with), was only partially addressed in the first review.

This is a major point, and it needs to be addressed before final publication. To be more precise: the discussion of the ability of the techniques to correctly measure under real-life conditions (line 388 ff) is essential, but still too optimistic. For instance (cf. line 390 ff.), it is obvious that the results achieved for unseeded techniques show severe deficiencies for rain intensities higher than 30 mmh-1. This aspect however is qualified by underlining the advantages of the technique, e.g. not having to rely on artificial particles to be added to the flow, etc. The doubtlessly existing advantages of unseeded techniques cannot hide the fact that overland flow velocities for higher rainfall intensities cannot be estimated accurately. On the other hand, statements like the one in line 394 is rather wishful thinking, relating to other studies which may show the potential under real-life conditions, but for single events and without the interference of raindrops. IMO the study here provides evidence that the evaluated techniques have limitations, and this should be unambiguously stated.

In a similar fashion, in the Conclusions (line 444, ff) it should be made clearer which method can potentially be applied under which conditions (shallow flows, events of low intensity, influence of raindrops for different intensities). In the current version of the manuscript, I am missing clear sentences like "seeded techniques are not able to measure velocities in areas with extreme shallow flows" which are included in the answer to previous reviews. I think making these aspects crystal-clear (e.g. like "method xy is suitable for rain intensities below a threshold of yz mm h-1") does not diminish the value of this excellent work, but it helps to do further research or to apply techniques under real-life conditions.

**Response:**

Following the comments of the reviewer, we have tried to un-bias the discussion and conclusions of the manuscript to reflect the remaining challenges and the how to deal with some limitations of imaging techniques during rainfall events in urban catchments. Thus, the first paragraph of the discussion (section 3.3) has been rewritten clarifying the limitations of unseeded techniques as follows (Pg 19, Ln 404):

*"The assessment of different imaging velocimetry techniques and the analysis of the influence of different factors on the velocity results contribute to understanding how these methodologies could be adequately transferred to real urban catchments. The use of these techniques would favor new velocity data sources to calibrate physically-based urban drainage models, such as traffic, public or surveillance cameras (Leitão et al., 2018; Moy de Vitry et al. 2020) or even unmanned aerial vehicles, which have already been used in river flow measurements (e.g. Lewis and Rhoads, 2018; Pearce et al., 2020). The insights gained in this study show the limitations of unseeded LSPIVu and BIV techniques to estimate runoff velocities under high rain intensity conditions or when complex flows are developed. Raindrop impacts on the water surface produce disturbances in the movement of the bubbles used as tracers that can prevent cross-correlation algorithms from obtaining reliable velocity distributions. In our experiments, this problem was observed when the rain intensity was higher than 30 mm/h, but this threshold may vary depending on the overland flow velocity or the raindrop kinetic energy. However, as there is no need to add artificial particles in the unseeded techniques they benefit from being straightforward to implement. Their ability to estimate velocities in extremely shallow flows,*

*where particles tend to be deposited, also make these techniques a potential tool for measuring velocities in field applications without the interference of raindrops or under light rain conditions. In contrast, it was observed that using artificial particles as tracers makes the LSPIV and LSPIVb techniques robust against heavy rain conditions and complex flows, such as those developed in Area 3 of the present study (Fig. 8). Therefore, the use of seeded techniques is recommended to estimate overland velocities in real urban catchments under rainy conditions, or when the measured flows are not simple enough. Special attention must be paid to the deposition of particles when the flow is extremely shallow."*

In addition, the second and third bullet of conclusions have been modified to include clearer sentences and specify which method can potentially be applied under which conditions (Pg 22, Ln 467):

*"- Both the seeded and unseeded techniques provide suitable velocity distributions in cases of unidirectional flows and the lowest rain intensity of 30 mm/h, with an offset of approximately 0.05 m s-1 between them. This offset is a consequence of the different tracers used in the seeded and unseeded experiments, which are affected to different degrees by raindrop impacts and may be transported at different velocities. Lower velocity indexes are thus required in the case of unseeded techniques to convert the results to depth-averaged velocities and these are affected by rain intensity.*

*- LSPIVu and BIV unseeded techniques are not able to estimate runoff velocities for higher rain intensities due to the disturbances introduced by raindrop impacts, which prevent cross-correlation algorithms from obtaining displacements and thus velocity distributions. The use of artificial particles as tracers by LSPIV and LSPIVb makes these seeded techniques robust against heavy rain conditions and are thus recommended in future field studies during rain events. Seeded techniques are also able to measure complex flows, where bubbles have difficulties in following the overland flow avoiding unseeded techniques to determine velocities. However, unseeded techniques can be suitable in field and laboratory applications in unidirectional flows and without the interference of raindrops or under light rain conditions, since they require a simpler experimental setup and are able to measure velocities in extremely shallow waters where artificial particles tend to be deposited."*

**R3C2**: line 19 - replace "feasibility" with "potential". The study illustrates the feasibility for controlled experiments but not for real-life applications!

**Response:**

This is completely true, so it was changed (Pg. 1 Ln. 19). Following this comment, "feasibility" has been also substituted by "potential" in introduction (Pg. 3 Ln. 8) and discussion (Pg. 20 Ln. 427).

**R3C3:** Abstract as well as Conclusion: despite it is mentioned in the Conclusion, that further research in real urban catchments should be done: it must be emphasized that the usefulness, i.e. the robustness of the techniques for real-life applications yet remains to be proven by means of further studies in non-controlled environments. With this respect, tangible recommendations on how to approach a validation under real-life conditions should be made to facilitate future research. What about the influence of

*different surfaces/surface roughness in real-life catchments, or the influence of wind, similarly occurring as raindrops, etc.*

**Response:**

We agree with the reviewer in the importance of emphasizing this point. The following sentence has been added to the abstract (Pg. 1 Ln. 21):

*"The robustness of the techniques for real-life applications yet remains to be proven by means of further studies in non-controlled environments."*

In addition, a comment regarding further research on non-controlled conditions has been added to conclusions (Pg. 23 Ln. 501):

*"The potential use of seeded and unseeded techniques in urban catchments has been proven, but future research should be oriented towards studying their robustness in real-world applications under non-controlled environments. The influence of the wind on rainfall distribution, catchment surface roughness or variable illumination conditions should be assessed in order to develop suitable pre-and post-processing procedures and correctly estimate runoff velocity results."*

Other minor issues:

- line 120: replace "rugosity" with "roughness". Plus, it would be very helpful to know the roughness value of the concrete slab in the physical model to put results into context, e.g. when considering other surfaces.

**Response:**

Fixed (Pg 5, Ln 126). The roughness value following Naves et al. (2019a) is 0.016. It has been added to the facility description in Pg 3, Ln 94. It is true that this value can be helpful if the present results are compared with works on other surfaces, thank you for the remark.

*"The roughness value of the roadway concrete surface is 0.016 (Naves et al., 2019a)."*

- line 288: "To do this and following the previous results". Awkward formulation. Consider rewriting.

**Response:**

We have rewritten it for a better readability as follows (Pg 12, Ln 289):

*"Following the previous results (Sect. 3.1), the reference values of the parameters (Table 1) have been considered for this comparison."*

- The English should be checked again prior final publication! Sentences are partially very long, i.e. complex (e.g. line 301 ff). Along with the language check, the authors may want to simplify the text at a few occasions in order to improve readability.

**Response:**

We have revised the full text trying to improve readability. In addition, the new version of the manuscript has been again checked by a native professional to ensure the quality of the writing. The changes are spread through the entire manuscript.